# SEARCHING META REASONING SKELETON TO GUIDE LLM REASONING

## ABSTRACT

Meta reasoning behaviors work as a skeleton to guide large language model (LLM) reasoning, thus help to improve reasoning performance. However, prior researches implement meta reasoning skeleton with manually designed structure, limiting ability to adapt to query-specific requirement and capture intricate logical dependency among reasoning steps. To deal with the challenges, we represent meta reasoning skeleton with directed acyclic graph (DAG) to unify skeletons proposed in prior works and model intricate logical dependency. Then we propose AutoMR, a framework that searches for query-aware meta reasoning skeleton automatically inspired by automated machine learning (AutoML). Specifically, we construct search space based on DAG representation of skeleton and then formulate the search problem. We design a dynamic skeleton sampling algorithm by expanding meta reasoning skeleton along with reasoning context at inference time. This algorithm can derive any meta reasoning skeleton in search space efficiently and adapt skeleton to evolving base reasoning context, thus enable efficient query-aware skeleton search. We conduct experiments on extensive benchmark datasets. Experimental results show that AutoMR achieves better reasoning performance than previous works broadly.

## 1 INTRODUCTION

Large language model (LLM) demonstrate superior performance on complex tasks such as math Q&A when equipped with step-by-step reasoning ability (Wei et al., 2022; OpenAI, 2024; DeepSeek-AI, 2025). Researches on cognition divide reasoning into two levels: base reasoning (reasoning for problem directly) and meta reasoning (higher-level reasoning about how to reason) (Flavell, 1979). Meta reasoning, considered a unique ability of human cognition (Ackerman & Thompson, 2017), entails awareness of one's reasoning process and the deliberate selection of reasoning strategies. For instance, when encountering difficulty with math problem, humans shift solution by thinking "This approach is not working; I should try another method..." or they may verify their reasoning steps by reflecting "Some steps may have errors. Let me check a previous step..." These behaviors do not directly solve the problem itself but instead organized as skeleton to guide the reasoning process.

Inspired by such human behaviors, previous studies proposed to incorporate meta reasoning into LLM to guide their reasoning process and thereby enhance performance on complex reasoning tasks (Gao et al., 2024; Qi et al., 2025; Sui et al., 2025; Liu et al., 2025). Recent approaches typically predefine a set of meta reasoning strategies for intermediate reasoning steps and employ manually designed structures (e.g. sequential, parallel and tree) to organize the strategies into meta reasoning skeleton. For example, rStar (Qi et al., 2025) and Meta-Reasoner (Sui et al., 2025) both define stepwise strategies such as decomposing question into sub-questions. rStar leverages Monte Carlo Tree Search (MCTS) (Coulom, 2006) to select and organize strategies, whereas Meta-Reasoner arranges them in a sequential way and selects at each step via multi-armed bandit (Gittins, 1979). An intuitive illustration of these manually designed skeleton is provided in Figure 2.

The aforementioned methods based on manually designed meta reasoning skeleton improved LLM reasoning performance. However, evidence from cognition science suggests that meta reasoning skeletons should vary for different queries, due to reasoner ability, query difficulty, discipline characteristic, etc. (Scott & Berman, 2013; Erickson & Heit, 2015; Rouault et al., 2018).

For example in Figure 1, knowledge-intensive problems (Q3 about biology) rely more heavily on knowledge-recall strategy while shallower thinking depth than thinking-intensive problem (Q1 and Q2 about math). More difficult problems (Q1) may demand more parallel reasoning branches with solution exploration strategy than easier one (Q2). Besides, the logical dependency of reasoning steps can be too intricate (Besta et al., 2024) to capture by sequential,

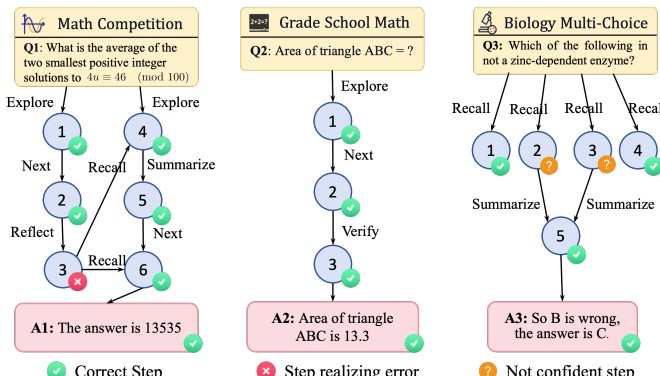

Figure 1: Human behaviors in meta reasoning for three questions about math (Q1 and Q2) and biology multi-choice (Q3).

parallel, or tree-structured skeletons in prior works. The skeleton of Q1 involves parallel branches (steps 1–3 forming one branch while steps 4–6 another) and multiple dependency (step 6 simultaneously depends on step 5 as well as step 3 from early branches). Skeleton of Q3 summarizes two steps (step 2 and 3) to make it answer confident. Apart from that, these manually designed skeletons have their inherent limitations. For example, sequential skeleton forces each step to attend to all previous steps, which can introduce hallucinations and error accumulation (Muennighoff et al., 2025; Sui et al., 2025). While parallel or tree structures isolate branches (Brown et al., 2024), preventing cross-branch information flow and leading to redundant reasoning steps that do not contribute to the final answer. The query-specific requirement and the intricate logical dependency among reasoning steps make it challenging for existing methods with limited manually designed meta reasoning skeletons (Figure 2) to work well across all queries.

Automated machine learning (AutoML) seeks to generate machine learning configurations for given task in a data-driven manner (Shen et al., 2024), thereby reducing the need for manual design and tuning for neural architectures (Elsken et al., 2019) and hyperparameter (Feurer & Hutter, 2019). Inspired by success of AutoML, we propose AutoMR, a framework that automatically searches for query-aware meta reasoning skeletons to guide LLM to reason for correct answer. In this framework, we represent meta reasoning skeleton as single-source edge-heterogeneous directed acyclic graph (DAG) because it can cover skeleton in prior works, capture intricate logical dependencies and address inherent issues of manually designed skeletons. Specifically, we first design an extensive DAG-based skeleton search space. Then we formulate the meta reasoning skeleton search problem, which poses two technical difficulties specific to query-aware skeleton search. The first is to derive any skeleton for given query from the extensive search space efficiently. The other is to adapt derived skeleton to evolving base reasoning context, considering inherent step-by-step property of reasoning process. To tackle the difficulties, we design a skeleton sampling algorithm that expands meta reasoning skeleton node by node dynamically based on base reasoning context at inference time. We prove that this algorithm introduces minimal additional computation overhead compared with naive LLM reasoning process. Compared with prior meta reasoning method, our search for meta reasoning skeleton improves reasoning performance. Moreover, we show that our search and inference algorithm is efficient theoretically and empirically.

We summarize our contributions as follows:

- We propose AutoMR to search for query-aware meta reasoning skeleton, where we represent meta reasoning skeleton as DAG to capture intricate logical dependency among reasoning steps.

- We design an extensive skeleton search space based on DAG. Additionally, we introduce an dynamic skeleton sampling algorithm that can derive any skeleton in search space efficiently and adapt skeleton to evolving base reasoning context at inference time.

- We conduct experiments on benchmark datasets across different disciplines and difficulties. Experimental results show that AutoMR demonstrates better reasoning performance than previous meta reasoning methods, with high search and inference efficiency.

## 2 RELATED WORKS

**Meta Reasoning in LLM.** Meta reasoning is an ability of human cognition involving determining reasoning strategy about how to reason (Flavell, 1979; Ackerman & Thompson, 2017). Previous works explored to introduce meta reasoning into LLM to guide it reasoning (Liu et al., 2025; Alazraki & Rei, 2025; Yan et al., 2025; Xiang et al., 2025; De Sabbata et al., 2024; Wan et al., 2025; Didolkar et al., 2024). Meta Reasoning Prompt (MRP) (Gao et al., 2024) includes classic strategies like CoT (Wei et al., 2022), Self-Refine (Madaan et al., 2023), etc. It first prompts LLM to choose one strategy for given query and then reason guided by that strategy. Strategies in MRP are holistic, meaning that MRP uses only one strategy for the whole reasoning process without adjusting when reasoning progressing. In contrast, recent methods usually use step-wise meta reasoning strategies (Yang et al., 2025a;b) and choose strategy for each step during reasoning. For example, rStar (Qi et al., 2025) define step-wise reasoning strategies such as proposing a sub-question, and then use MCTS to build tree-structured meta reasoning skeleton. Meta-Reasoner (Sui et al., 2025) also uses step-wise reasoning strategies but organizes them with sequential skeleton and uses multi-armed bandit to select strategy for each step. This kind of methods incorporate more fine-grained meta reasoning guidance and allow adjusting strategies during reasoning, thus performing better empirically than MRP.

**Automated Machine Learning (AutoML).** AutoML aims to search for high-performing machine learning (ML) configuration for given task automatically, reducing demand for human manual design (He et al., 2021) to adapt to task-specific requirement. Typical AutoML atomizes ML configurations to construct search space and develop search algorithm to find effective candidates (Shen et al., 2024). Previous works implemented this idea for multiple ML configurations such as neural architecture search (NAS) (White et al., 2023; Liu et al., 2019; Pham et al., 2018) and hyperparameter search (Yang & Shami, 2020; Shen et al., 2023), and have achieved success. For example, architectures found by NAS surpass human-designed ones on various tasks, such as computer vision (Real et al., 2019) and natural language processing (So et al., 2019). Recent works explored integrating AutoML with LLMs, like automating LLM agent workflow building (Zhuge et al., 2024; Zhang et al., 2025a; Saad-Falcon et al., 2025). However, applying AutoML method to search for meta reasoning skeleton is non-trivial due to factors specific to LLM reasoning task, including query-specific requirement, intricate logical dependency, and evolving reasoning context.

## 3 PROPOSED METHOD

We introduce AutoMR that automatically searches for query-aware meta-reasoning skeletons to guide LLM reasoning. Section 3.1 presents a unified perspective on meta-reasoning skeleton in existing meta-reasoning methods based on DAG to capture intricate logical dependency. With this unified view, we construct our skeleton search space. Section 3.2 formulates the meta-reasoning skeleton search problem and details our overall search strategy. Finally, Section 3.3 discusses comparison with techniques in AutoML and analyzes our advantage specific to LLM reasoning tasks.

### 3.1 SEARCH SPACE

Given a query $q$, let $\mathcal{S}$ denote the set of meta reasoning strategies for intermediate reasoning steps. The objective of a meta-reasoning method is to organize strategies from $\mathcal{S}$ into meta reasoning skeleton to direct LLM on performing reasoning to answer $q$.

Prior works use manually designed meta reasoning skeleton structure (e.g. sequential, parallel, tree-structured in Figure 2). To unify these designs and capture intricate logical dependencies (Figure 1), we represent meta reasoning skeleton as a *single-source, edge-heterogeneous directed acyclic graph (DAG)*. Formally, a meta reasoning skeleton can be represented as a DAG $\alpha = (\mathcal{V}, \mathcal{E}, \tau, \mathcal{S})$. Node $n_i = (i, c_i) \in \mathcal{V}$ representing a reasoning step, $i$ being the topological index and $c_i$ textual content of the step. Edge $(i, j) \in \mathcal{E}$ indicating reasoning progression from $n_i$ to $n_j$. $\tau : \mathcal{E} \to \mathcal{S}$ maps edge to its strategy, under which LLM generates the reasoning text. There exists a unique source node $n_0$ with $c_0 = q$, making $\alpha$ single-source. With above representation, we have Proposition 1 to cover the skeletons in prior works. See Appendix B.1 for proof.

**Proposition 1.** *Sequential, parallel, and tree structured skeletons can all be represented as single-source, edge-heterogeneous DAGs.*

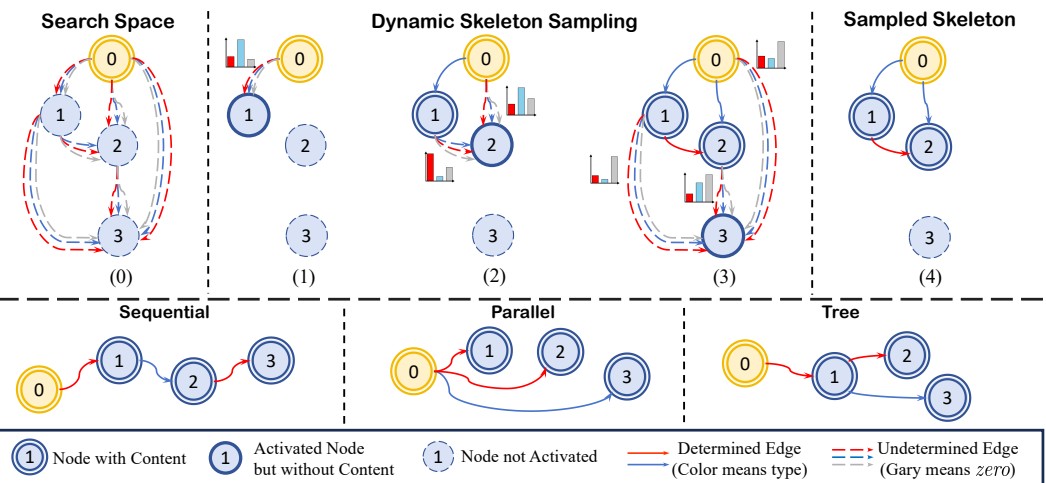

Figure 2: Overview of the AutoMR. **Top:** Illustration of search space, an example skeleton sampling process and resulting sampled skeleton. Node 0 is the single source node representing query. Steps (1)(2)(3) show how nodes 1, 2, and 3 are successively added to partial skeleton. For clarity, we display only 4 nodes and 2 types of meta reasoning strategies (red and blue edges), and the *zero* option (gray edges); In practice, the number of nodes can be arbitrary if token budget is satisfied and we actually implement richer strategies. **Bottom:** Search space subsumes sequential, parallel, and tree-structured skeletons.

Based on this unified view, we construct search space to contain all skeletons represented by single-source edge-heterogeneous DAG as shown in Figure 2, as long as the sum of tokens for all node content except source node (i.e. number of tokens generate by LLM) dose not reach token budget $\mathcal{B}$, where $\mathcal{B}$ is a hyperparameter.

We summarize the meta reasoning behaviors in previous works about LLM reasoning (Gandhi et al., 2025; Chen et al., 2025), which gives meta reasoning strategy set $\mathcal{S} = \{$**Next**, **Reflect**, **Explore**, **Decompose**, **Summarize**, **Recall**, **Answer**$\}$. All of these meta reasoning strategies are implemented by designed prompt. Functions and prompt of these strategies are summarized in Table 3 in Appendix A.1. Following previous works (Liu et al., 2019), we also introduce a special *zero* edge type to indicate an edge in fact dose not exists.

Given meta reasoning strategy set $\mathcal{S}$ and token budget $\mathcal{B}$, search space $\mathcal{A}$ is defined as follows,

$$\mathcal{A} = \Big\{ \alpha = (\mathcal{V}, \mathcal{E}, \tau, \mathcal{S}) \mid \alpha \text{ is single-source DAG}, \ \tau : \mathcal{E} \to \mathcal{S}, \ \sum_{n_i \in \mathcal{V} \setminus \{n_0\}} |c_i| \leq \mathcal{B} \Big\}, \quad (1)$$

where $\mathcal{V} \setminus \{n_0\}$ is node set without $n_0$ and $|c_i|$ denote number of tokens in content $c_i$. As illustrated in Figure 2 (bottom), this search space includes all single-source DAGs, thus subsuming skeletons considered in prior meta-reasoning methods, such as sequential, parallel, and tree-structured forms.

## 3.2 SEARCH STRATEGY

Next, we now provide the formal definition of *meta-reasoning skeleton search problem*. Considering that the meta-reasoning skeleton should depend on the specific query (e.g., query difficulties and discipline characteristics), the problem is formulated as follows.

**Definition 1** (Meta-Reasoning Skeleton Search Problem). *Let $\mathcal{S}$ denote meta reasoning strategy set and $\mathcal{A}$ the skeleton search space defined on $\mathcal{S}$. $(q, a)$ is query–answer pair from dataset $\mathcal{D}$. Given policy $P$ that derives a meta reasoning skeleton $\alpha_q \in \mathcal{A}$ for query $q$, the search objective is*

$$\arg\max_P \mathbb{E}_{(q,a) \sim \mathcal{D}, \alpha_q \sim P(\cdot|q)} [r(a, LLM(q; \alpha_q))]. \quad (2)$$

*Here $LLM(q; \alpha_q)$ denotes LLM reasoning on query $q$ under guidance of $\alpha_q$, and $r$ measures reasoning performance against the ground-truth answer $a$.*

When implementing a policy $P$ for deriving a query-aware skeleton, this search problem poses two technical challenges specific to LLM reasoning. **First,** the search space is extensive, so the

derivation procedure must efficiently explore it to recover arbitrary skeletons in it. **Second**, because reasoning process unfolds step by step (Wei et al., 2022; Nye et al., 2021), the derivation process should adapt meta reasoning strategy at each step in skeleton to evolving base reasoning context, rather than fixing the skeleton a priori before reasoning for given query.

To address above difficulties, Section 3.2.1 introduces a skeleton-sampling algorithm that expand skeleton node by node dynamically, along with base reasoning context at inference time. We prove that the algorithm can cover any skeleton in search space within minimal additional computation compared with naive LLM reasoning process; Section 3.2.2 presents the overall search algorithm.

### 3.2.1 DYNAMIC SKELETON SAMPLING AT INFERENCE TIME

We introduce an efficient algorithm that sample DAG-based skeleton dynamically to implement policy $P(\cdot \mid q)$. Common search methods for DAG fix the number of nodes and optimizing all edge probabilities such as DARTS (Liu et al., 2019) for neural architecture search or GPTSwarm (Zhuge et al., 2024) for multi-agent workflow. However, these techniques are unsuitable for search meta reasoning skeleton, because they contradict the step-by-step unfolding nature of reasoning process. Considering step-by-step nature of reasoning, step-wise meta reasoning strategy should adapt to *current* base reasoning context. This makes it necessary to *interleave* meta reasoning with base reasoning.

To realize this, we sample skeleton starting from the single source node as a partial skeleton, and then expand it node by node in topological order, dynamically aligning with step-by-step base reasoning at inference time .

Specifically, we set content $c_0$ of $n_0$ as $q$, forming a *partial* architecture. Expansion then proceeds in topological order. For each target node $n_i$, we determine the existence and types of incoming edges before (optionally) generating its content. Concretely, when visiting $n_i$ we first *activate* it (no content yet) and perform following three steps.

**Step1: Determine incoming edges for meta reasoning.** Traverse existing nodes $n_j$ ($0 \leq j \leq i-1$) in reverse order (from $n_{i-1}$ to $n_0$) and sample a strategy $s_{(j,i)} \in \mathcal{S} \cup \{zero\}$ for each potential edge $(j, i)$. Each sampling is conditioned on the predecessor content $c_j$, the already chosen strategies $s_{(>j,i)}$ for $n_i$, and the current base rea-

---

**Algorithm 1** Dynamic Skeleton Sampling at inference time

---

**Require:** Query $q$, token budget $\mathcal{B}$
**Ensure:** Meta reasoning architecture $\alpha_q$
1: Initialize $\alpha_q$ as empty DAG, $i \leftarrow 0$
2: **while** $\mathcal{B}$ is not reached **do**
3:    **for** $j$ from $i$-1 to 0 **do**
4:       Sample $s_{(j,i)} \sim p_\theta(s_{(j,i)}|c_j, s_{(>j,i)}, c_{:i-1})$ with MLP
5:    **end for**
6:    **if** all sampled strategies are *zero* **then**
7:       Generate final answer and **return**
8:    **end if**
9:    Generate content $c_i$ for $n_i$, $i \leftarrow i + 1$
10: **end while**
11: Generate final answer

---

soning context $c_{:i-1}$ (the contents of $n_0, \ldots, n_{i-1}$), which is computed as $p(s_{(j,i)}|c_j, s_{(>j,i)}, c_{:i-1})$.

**Step2: Check completion.** If all sampled strategies are *zero* (no edge enters $n_i$), we deem the skeleton complete without adding $n_i$ and prompt the LLM to produce the final answer from the current context $c_{:i-1}$.

**Step3: Generate base reasoning content.** If at least one incoming edge exists, we prompt the LLM *under the guidance* of the sampled strategies $s_{(<i,i)}$ (excluding *zero*) and the contents of $n_i$'s predecessors to produce the next base reasoning step; the generated text is assigned to $c_i$, and $n_i$ (with its incoming edges) is added as a *node with content*.

Then we repeat this expansion for $n_{i+1}$ until Step 2 triggers or token budget is reached.

We implement $p(\cdot)$ with a multi-layer perception (MLP) parameterized with $\theta$. The MLP takes representations of $c_j$, $s_{(>j,i)}$, and $c_{:i-1}$ as input and outputs logits followed by softmax to obtain distribution over $\mathcal{S} \cup \{zero\}$. These representations are cached byproducts of the ongoing LLM inference (i.e. pooled hidden states), thus requiring no additional LLM calls. If the sampled skeleton $\alpha_q$ contains $|\mathcal{V}|$ nodes, its policy (also parameterized with $\theta$ now) log-probability factorizes as

$$\log P_\theta(\alpha_q|q) = \sum_{i=1}^{|\mathcal{V}|-1} \sum_{j=0}^{i-1} \log p_\theta(s_{(j,i)}|c_j, s_{(>j,i)}, c_{:i-1}). \tag{3}$$

The sampling process is shown in Figure 2 and formalized in Algorithm 1. According to Algorithm 1, meta reasoning strategy sampling is conditioned on *current* base reasoning context at each step, thereby yielding a query-aware architecture since reasoning context traces back to $c_0 = q$. Implementation details are in Appendix A.3. For Algorithm 1, we have Proposition 2.

**Proposition 2.** *Algorithm 1 can derive any $\alpha \in \mathcal{A}$, within $O(|\mathcal{V}|^2)$ additional MLP calls (line4) compared with naive LLM reasoning process.*

The time complexity of naive LLM reasoning process is proportional to $\mathcal{B}^2$. But $|\mathcal{V}| \ll \mathcal{B}$ because one step usually contains many tokens, and MLP uses much less computation than LLM, so AutoMR introduces minimal additional computation relative to naive LLM reasoning. We provide proof of Proposition 2 and detailed efficiency analysis in Appendix B.2.

### 3.2.2 OVERALL SEARCH ALGORITHM

With $P_\theta(\alpha_q|q)$ defined in (3), we follow REINFORCE (Williams, 1992; Zoph & Le, 2017), a policy gradient algorithm implementing unbiased empirical approximation of objective, to optimize $\theta$. Specifically, we sample batches with $N$ query-answer pairs $(q_i, a_i)$ from training set each time and optimize $\theta$ with these batches iteratively. For each $(q_i, a_i)$ in batch, we sample $M$ skeletons $\alpha_{q_i}^j$ from $P_\theta(\cdot|q_i)$ and evaluate their performance with $r(\cdot)$ respectively. The update to $\theta$ in each iteration by estimated policy gradient with a batch is as follows, where $\eta$ is learning rate:

$$\theta \leftarrow \theta + \frac{\eta}{MN} \sum_{i=1}^{N} \sum_{j=1}^{M} [r(a_i, \text{LLM}(q_i, \alpha_{q_i}^j)) \nabla_\theta \log P_\theta(\alpha_{q_i}^j|q_i)]. \tag{4}$$

The overall search algorithm and implementation is provided in Appendix A.4. We do not tune LLM parameters directly, thus enabling efficient search. For inference, we follow Algorithm 1 for each query to sample meta reasoning skeleton, generate base reasoning and output final answer.

### 3.3 TECHNICAL COMPARISON WITH AUTOML

Different from prior meta reasoning methods that rely on manually designed skeleton (Qi et al., 2025; Sui et al., 2025), AutoMR draws inspiration from AutoML to search for query-aware meta reasoning skeleton from DAG-based search space, thereby addressing query-specific requirements. Technically, AutoMR is related to topics in AutoML such as neural architecture search. Recent studies have extended AutoML ideas to LLM-related tasks, such as automating agent workflow building (Zhuge et al., 2024; Zhang et al., 2025a). However, the unique properties of LLM reasoning tasks make AutoMR particularly suited for meta reasoning skeleton search. First, reasoning queries often exhibit highly specific demands, making a single meta reasoning skeleton insufficient. Second, the reasoning process typically involves intricate logical dependencies. Third, reasoning unfolds step by step, with the base reasoning context dynamically evolving as each new step is generated. These characteristic fundamentally differs from those of neural architecture or agent workflow, which is usually fixed for all queries or static during inference. For example, Prior approaches (Zoph et al., 2018; Zhuge et al., 2024) generally output a single architecture or agent workflow for all queries. While instance-aware methods (Cheng et al., 2020; Zhang et al., 2025a) produce input-specific architecture or workflow that remain static during inference. Such differences in task properties makes the search techniques in these methods perform well in their target scenarios but cannot be applied to meta reasoning skeleton search directly. We compare these search techniques empirically by ablation study in Section 4.3.

## 4 EXPERIMENTS

### 4.1 SETUP

**Baselines.** We implement the following types of baselines: **(1) Classic methods**, including **Direct-I/O** and **CoT** (Wei et al., 2022). **(2) Meta reasoning methods**, including **MRP** (Gao et al., 2024), **rStar** (Qi et al., 2025) and **Meta-Reasoner** (Sui et al., 2025). We also include **MaAS** (Zhang et al., 2025a), a method using NAS technique to automate multi-agent workflow building.

AutoMR and all the baselines are implemented based on two LLMs including LLaMA3.2-3B-Inst (hereinafter referred to as "LLaMA") (Meta-AI, 2024) and Qwen2.5-3B-Inst (hereinafter referred to as

Table 1: The overall performance on math Q&A and general multi-choice. Letters after method names means the used skeleton structure. S: Sequential; T: Tree; G: DAG; "-" means not applicable.

| Method | MATH-500 | | GSM8K | | AMC | | Olympiad | |
|---|---|---|---|---|---|---|---|---|
| | LLaMA | Qwen | LLaMA | Qwen | LLaMA | Qwen | LLaMA | Qwen |
| Direct-I/O (-) | 12.6 | 16.8 | 11.1 | 15.8 | 12.0 | 8.4 | 3.7 | 5.5 |
| CoT (S) | 36.8 | 61.6 | 71.1 | 85.3 | 21.2 | 34.9 | 11.9 | 26.2 |
| MRP (-) | 40.8 | 63.8 | 74.6 | 88.2 | 25.3 | 33.7 | 11.6 | 26.6 |
| Meta-Reasoner(S) | 44.4 | 65.4 | 76.8 | 87.0 | 26.5 | 36.1 | 13.1 | 27.4 |
| rStar (T) | 46.6 | 67.0 | 78.9 | 88.7 | 15.7 | 32.5 | 15.1 | 25.4 |
| MaAS (S) | 46.2 | 63.6 | 76.4 | 86.4 | 24.1 | 33.7 | 12.6 | 27.7 |
| AutoMR (G) | **50.2**$_{\uparrow 13.4}$ | **69.6**$_{\uparrow 8.0}$ | **81.9**$_{\uparrow 10.8}$ | **91.5**$_{\uparrow 6.2}$ | **30.1**$_{\uparrow 8.9}$ | **38.6**$_{\uparrow 3.7}$ | **17.4**$_{\uparrow 5.5}$ | **30.4**$_{\uparrow 4.2}$ |

| Method | Science | | Humanities | | Social | | Other | |
|---|---|---|---|---|---|---|---|---|
| | LLaMA | Qwen | LLaMA | Qwen | LLaMA | Qwen | LLaMA | Qwen |
| Direct-I/O (-) | 16.3 | 32.7 | 11.5 | 25.1 | 15.8 | 39.0 | 14.5 | 29.1 |
| CoT (S) | 31.5 | 41.6 | 22.4 | 28.3 | 37.3 | 51.5 | 31.3 | 39.8 |
| MRP (-) | 36.4 | 42.8 | 24.2 | 30.1 | 40.6 | 53.5 | 32.8 | 41.6 |
| Meta-Reasoner (S) | 44.3 | 45.4 | 30.6 | 31.9 | 47.2 | 55.0 | 36.4 | 42.2 |
| rStar (T) | 42.6 | 43.6 | 30.0 | 30.8 | 46.8 | 55.4 | 34.8 | 36.0 |
| MaAS (S) | 44.6 | 45.5 | 29.7 | 31.0 | 46.2 | 56.0 | 35.6 | 41.7 |
| AutoMR (G) | **48.9**$_{\uparrow 17.4}$ | **49.4**$_{\uparrow 7.8}$ | **33.2**$_{\uparrow 10.8}$ | **33.7**$_{\uparrow 5.4}$ | **51.0**$_{\uparrow 12.7}$ | **57.4**$_{\uparrow 5.9}$ | **38.8**$_{\uparrow 7.5}$ | **45.6**$_{\uparrow 5.8}$ |

as "Qwen") (Qwen-Team, 2025) to avoid impact on experimental results caused by unique properties of specific LLM (Gandhi et al., 2025). We set the same token budget to 1024 for all methods to ensure fair comparison. More implementation details of the baselines are introduced in Appendix C.1.

**Datasets and Metric.** We evaluate AutoMR and baselines on two domains, i.e. math Q&A and general multiple-choice. For math Q&A, we choose **GSM8K** (Cobbe et al., 2021), **MATH-500** (Hendrycks et al., 2021), **AMC** (including AMC 2022 and AMC 2023) and **Olympiad** (only open-ended text-only math subset to avoid influence from multi-modal and multilingual input information) (He et al., 2024) to evaluate. We use training split of **MATH** dataset to train AutoMR and baselines that need training. For general multiple-choice, we choose **MMLU-Pro** (Wang et al., 2024) and split it into four subsets as **Science**, **Humanities**, **Social** and **Other** referring to Zhang et al. (2025b), to evaluate. We collect training split of MMLU-Pro to train. Details of these datasets are summarized in Appendix C.2. We use **Accuracy** as metric to evaluate these methods .

## 4.2 PERFORMANCE COMPARISON

We report the overall performance of AutoMR and baselines on math Q&A datasets and general multiple-choice datasets (Table 1). Across both domains and model backbones, AutoMR consistently achieves the best results, highlighting its broad effectiveness. Our findings can be summarized as follows: (1). Effectiveness of meta reasoning methods. Meta reasoning approaches (MRP, Meta-Reasoner, rStar, and AutoMR) consistently outperform the standard CoT baseline. Notably, Meta-Reasoner—despite adopting the same sequential organization as CoT—achieves a substantial improvement, underscoring the benefits of incorporating meta reasoning behaviors. (2). Importance of fine-grained meta reasoning strategies. Among meta reasoning methods, those that leverage strategies for guiding intermediate reasoning steps (Meta-Reasoner, rStar, and AutoMR) outperform MRP, which relies on holistic strategy. This result highlights the advantage of fine-grained meta-level guidance during reasoning. (3). Advantage of DAG-based search space. Compared with Meta-Reasoner and rStar, which rely on manually designed sequential and tree-structured skeleton respectively, AutoMR achieves superior performance. (4). AutoMR surpasses automatic agent workflow MaAS, demonstrating that AutoMR is more proper for LLM reasoning tasks.

## 4.3 ABLATION STUDY

**Influence of token budget scaling.** Previous works shows that LLM reasoning performance improves whentoken budget increases (OpenAI, 2024; Snell et al., 2025). We evaluate the performance when scaling token budget $\mathcal{B}$. We compare AutoMR with baselines able to scale token

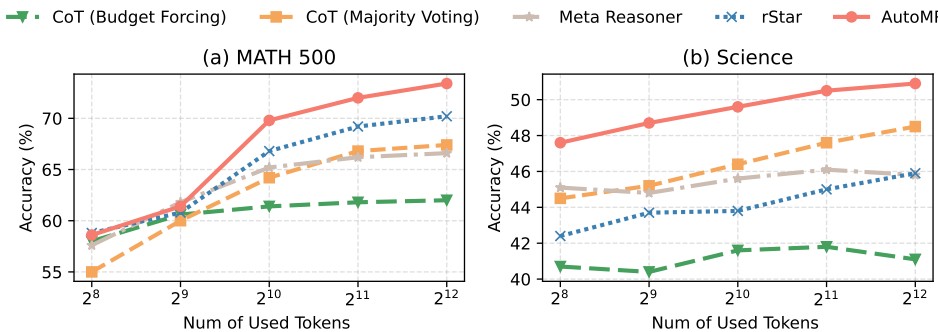

Figure 3: The scaling curve of AutoMR and baselines.

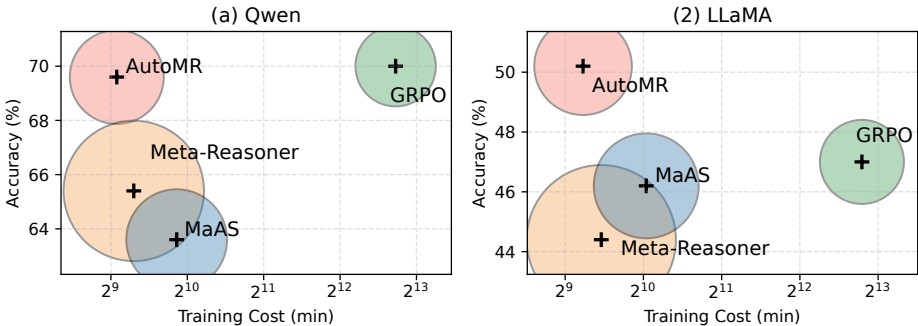

Figure 4: The training and inference cost and performance of AutoMR and baselines.

budget. Specifically, for CoT we implement sequential scaling technique **Budget Forcing** (Muennighoff et al., 2025) and parallel technique **Majority Voting** (Wang et al., 2023). We also choose **Meta-Reasoner** and **rStar** as baselines. We do not include **MaAS** as baselines to evaluate because it do not provide scaling technique in original paper. The scaling technique implementation details of these methods are in Appendix C.1. We evaluate on MATH-500 and Science based on Qwen. According to results in Figure 3, we observe that when token budget increases, each method improves performance on the whole. Specifically, the scaling efficiency on knowledge-intensive Science subset is much slower than that on thinking-intensive MATH-500, according with recent research (Zhao et al., 2025). Forcing sequential scaling (i.e. Budget Forcing and Meta-Reasoner) scale slowly. Majority Voting based on parallel skeleton and rStar based on tree-structured skeleton scale more efficiently than sequential ones. AutoMR achieve the highest scaling efficiency, because search space based on DAG in AutoMR allows more extensive skeleton exploration.

**Effectiveness of search strategy.** We evaluate the effectiveness of search strategy in Section 3.2 against **R**andom **S**earch (**RS**) (Bergstra & Bengio, 2012), a common AutoML baseline (Li & Talwalkar, 2020). We also assess effectiveness of dynamic skeleton sampling algorithm by comparing it with two variants. **Q**uery-**I**nvariant (**QI**), sampling single meta reasoning skeleton shared by all queries of a task, as in prior NAS methods (Liu et al., 2019; Pham et al., 2018). **C**omplete in **A**dvance (**CA**), sampling query-specific skeletons before reasoning starts but not based on reasoning context (Cheng et al., 2020; Zhang et al., 2025a). Implementation details of these sampling methods are in Appendix C.1. We compare them on MATH-500 and Science.

According to results in Table 2, AutoMR achieves the best performance compared with three variants, showing the effectiveness of proposed search strategy. In terms of skeleton sampling algorithm, AutoMR and CA both surpass QI, showing the importance of query-specific meta reasoning skeleton. Moreover, AutoMR performs better than CA, demonstrating the effectiveness dynamic skeleton sampling algorithm based on evolving reasoning context compared with the complete skeleton in advance.

Table 2: Ablation study on search strategy.

| Method | MATH-500 | | Science | |
|---|---|---|---|---|
| | LLaMA | Qwen | LLaMA | Qwen |
| RS | 36.2 | 59.4 | 38.5 | 43.3 |
| QI | 37.2 | 60.2 | 37.3 | 43.9 |
| CA | 50.0 | 66.2 | 45.7 | 47.1 |
| AutoMR | **50.2** | **69.6** | **48.9** | **49.4** |

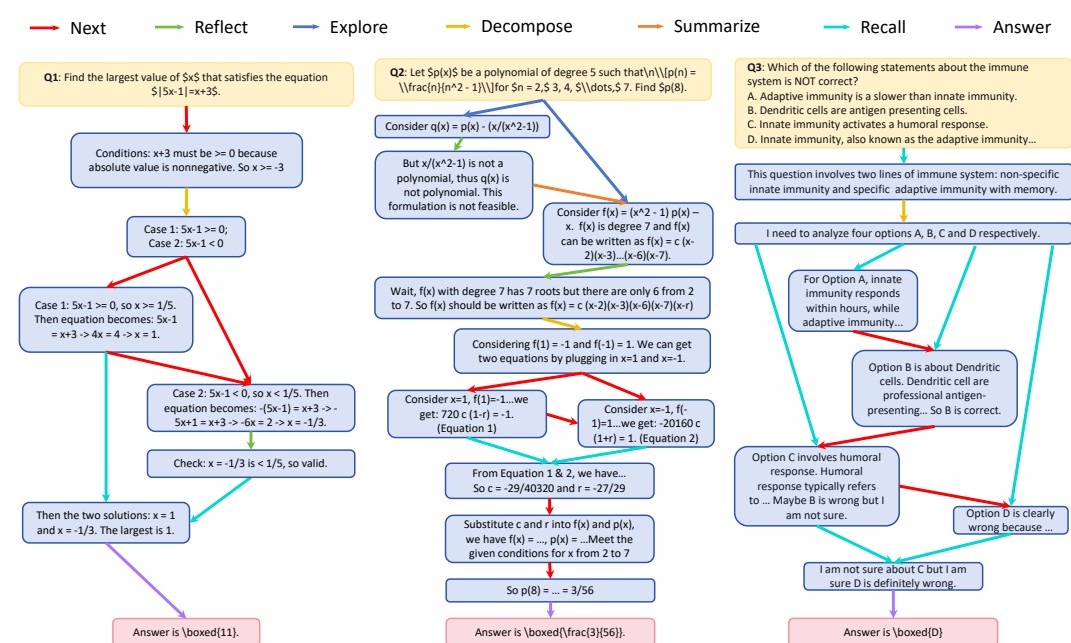

Figure 5: Searched skeletons for queries from MATH-500 Level1, Level5 and Science respectively.

**Training and inference efficiency.** To support theoretical analysis in Section 3.2.2 that AutoMR incurs minimal additional computation, we evaluate both training and inference costs of AutoMR and baselines requiring training, including **Meta-Reasoner** and **MaAS**, based on both Qwen and LLaMA on MATH-500 dataset. We also implement **GRPO** (Shao et al., 2024), a reinforcement learning method to enhance LLM reasoning, based on LoRA (Hu et al., 2022) as a baseline in our experiment setting. Results in Figure 4 show training cost (x-axis), performance on MATH-500 (y-axis), and inference cost (circle area). In terms of training, AutoMR and other two baselines require far less time than GRPO, which fine-tunes LLM parameters directly. However, only AutoMR achieves comparable performance with Qwen and even surpasses it with LLaMA. In terms of inference, AutoMR is slightly slower than naive reasoning process based on GRPO-trained LLM and slightly faster than MaAS, while being substantially more efficient than Meta-Reasoner, which relies on additional LLM calls to summarize reasoning progress. Instead, AutoMR employs a lightweight MLP to process representations produced during reasoning, avoiding extra LLM calls.

### 4.4 CASE STUDY

We visualize searched meta reasoning skeletons of three queries respectively in Figure 5. Q1 and Q2 come from MATH-500 while Q3 is from Science. According to three skeletons and their corresponding queries, we observe that AutoMR can search out query-aware skeleton, which is appropriate for given query considering query properties such as difficulty and discipline characteristics.

**Skeleton Cases of Queries from Different Tasks.** Q1 and Q2 correspond to math Q&A tasks, which are typically regarded as thinking-intensive, while Q3, drawn from the Science subset, concerns the history of biology and is considered knowledge-intensive. For two math queries, skeletons sampled by AutoMR exhibit deeper reasoning steps and employ more diverse meta reasoning strategies (e.g., Exploration and Reflection) than that sampled for Q3. By contrast, skeleton for Q3 emphasizes Recall strategy. This distinction aligns with the characteristics of thinking-intensive math versus knowledge-intensive history of biology.

**Skeleton Cases of Queries with Different Difficulties.** Both Q1 and Q2 are drawn from the MATH-500 dataset, Q2 belongs to the more challenging "Level-5" subset whereas Q1 comes from simpler "Level-1" subset. Correspondingly, skeleton for Q1 is more complex than that of Q2. In Figure 5, the skeleton for Q2 contains two reasoning branches, where the LLM explores two potential solutions, with the first attempt failing. It also incorporates Recall strategy to leverage intermediate result from

earlier steps. However, skeleton for simpler Q1 explores only single solution path, successfully solving the problem by that path and without recalling very early steps.

## 5 CONCLUSION

We propose AutoMR, a framework that searches for query-aware meta-reasoning skeleton to guide LLM reasoning. By formulating meta-reasoning as a search problem over DAG-based search space, AutoMR covers skeletons in prior works and can capture intricate logical dependencies among reasoning steps. AutoMR designs a dynamic skeleton sampling algorithm that can derive any skeleton in search space within minimal additional computation overhead, and make skeleton adaptable to evolving base reasoning context, thus enabling efficient search. Experiments on math Q&A and general multiple-choice benchmark datasets demonstrate consistent improvements over existing meta reasoning methods.

## REPRODUCIBILITY STATEMENT

We have made great efforts to to ensure reproducibility of our results. We give the implementation details of AutoMR and baselines in Appendix A and Appendix C.1. We open the source code of AutoMR with an anonymous repository as `https://anonymous.4open.science/r/Code-AutoMR-ED4C`.

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

# A    IMPLEMENTATION DETAILS

## A.1    META REASONING STRATEGY IMPLEMENTATION

The functions of meta reasoning strategies is summarized in Table 3. We design maybe more than one prompts for each strategy and sample one randomly when sampling strategy for an edge. Some prompts are used only for certain tasks and we indicate them in parentheses after the prompt. The prompts of all meta level strategies are as follows.

Table 3: Meta reasoning strategies.

| Strategy | Function |
|---|---|
| Next | Reason to next step. |
| Reflect | Reflect previous reasoning steps |
| Explore | Inspire divergent thinking |
| Decompose | Decompose current query and propose sub-question. |
| Summarize | Summarize previous reasoning steps. |
| Recall | Recall related knowledge or previous steps about problem. |
| Answer | Give answer and end current reasoning path. |

### Prompt for Next

- Next,
- Then,
- Now, let me move on to the next step.

### Prompt for Reflect

- Let me consider what part of the reasoning feels least certain, and how can it be examined.
- Wait, let me think if there anything missing in the current reasoning.
- Let me think does the current line of thought have any error.

### Prompt for Explore

- Let me consider which direction of thinking I should explore.
- Let me think what potential strategy has not yet been considered that could be the next solution path.
- Let me think what possible solution could be tried next.

### Prompt for Decompose

- This question is a bit complex, let me think how to decompose it into sub-questions that I can solve.
- The question feels too broad, let me think what smaller version could I tackle first.
- Let me think if I can express the problem in terms of simpler components or modules.
- Let me consider the options one by one. (General multi-choice)

> **Prompt for Summarize**
>
> - Let me summarize what have I established so far.
> - Let me summarize the current state of reasoning process, what's known, unknown, and assumed?
> - Let me consider if I can captures the essence of the reasoning so far with single sentence.

> **Prompt for Recall**
>
> - Let me think if I have encountered similar problems or if learned knowledge and previous intermediate step can be used here.
> - Let me think what prior reasoning steps are directly relevant here or this question connect to earlier results. (Math Q&A).
> - Let me recall which theorems, rules, or principles from earlier knowledge is related to this question. (General multi-choice).

> **Prompt for Answer**
>
> Let me give the answer according to current reasoning context.

## A.2   IMPLEMENTATION DETAILS FOR GENERATING BASE REASONING CONTENT

When generating base reasoning content for a node $n_i$, we use the following prompt template to build prompt for each incoming edge $(j, i)$ with strategy $s_{(j,i)}$ for $n_i$.

> **Prompt Template for Generating Base Reasoning Content**
>
> Let me attend to Step $\{j\}$, {prompt for strategy $s_{(j,i)}$}

If node $n_i$ has only one incoming edge, then we use the prompt for this edge to prompt LLM to generate base reasoning content $c_i$ for $n_i$. If the node $n_i$ has multiple incoming edges, we concatenate the prompts for all incoming edges into one prompt and use it to prompt LLM to generate base reasoning content $c_i$ for $n_i$.

## A.3   META REASONING STRATEGY SAMPLING

We implement an MLP model to sample strategy for edge $(j, i)$ from $n_j$ to $n_i$ by taking representations of potential predecessor node content $c_j$, already sampled strategy $s_{>j,i}$ and current base reasoning context composed of all node content in partial skeleton $c_{:i-1}$.

Specifically, we maintain a learnable embedding layer to map each strategy $s \in \mathcal{S} \cup \{zero\}$ to a dense embedding. For each node content $c$, we save the mean of "last hidden state" of the $c$ as semantic representation of the node content. "Last hidden state" is byproduct of LLM inference process for token distribution when generating each token, requiring no extra LLM invocation.

Finally, we build input for MLP according to $\mathrm{Concat}([e(c_j), \mathrm{Mean}(e(s_{>j,i})), \mathrm{Mean}(e(c_{:i-1}))])$, where $\mathrm{Concat}(\cdot)$ means concatenate vectors and $\mathrm{Mean}(\cdot)$ means calculate the mean of vectors. We use $\mathrm{Softmax}(\cdot)$ to process output of MLP and give the distribution of $s_{(j,i)} \in \mathcal{S} \cup \{zero\}$.

## A.4   OVERALL SEARCH ALGORITHM

We show the overall search algorithm in Algorithm 2.

The hyperparameters of the MLP implementation depend on the LLM we use. For an LLM with hidden size $d$, the input dim of our MLP is $3 * d$. Then we usually use a three-layer MLP that reduce

the dim to $2 \times d$, $d$, $d/2$ and finally to the number of meta reasoning behaviors. The MLP model is in fact a classification head that outputs the probability of each meta reasoning strategy.

We implement $N$ as 8, $M$ as 16 and learning rate $\eta$ to $5 \times 10^{-4}$ during search for both tasks. We refer to previous works (Zhuge et al., 2024; Zhang et al., 2025a; Xie et al., 2025; Hu et al., 2025; Cheng et al., 2020), implement techniques such as gradient clipping, to improve the stability and convergence rate of search algorithm. See our code for implementation details. We implement a rule-based $r$ by exactly matching final answer $\hat{a} = \text{LLM}(q, \alpha_q)$ given by LLM with ground-truth $a$ from dataset. Specifically,

$$r(a, \text{LLM}(q, \alpha_q)) = \begin{cases} 1, & \text{if } \text{LLM}(q, \alpha_q) = a, \\ -1, & \text{if } \text{LLM}(q, \alpha_q) \neq a. \end{cases}$$

---

**Algorithm 2** Overall Search Algorithm

---

**Require:** Dataset $\mathcal{D}$, learning rate $\eta$
**Ensure:** Trained $\theta$
1: Initialize $\theta$ randomly
2: **while** not convergence **do**
3:     Sample a batch $\{q_1, q_2, ..., q_N\}$ from $\mathcal{D}$
4:     Sample $\{\alpha_{q_i}^1, \alpha_{q_i}^2, ..., \alpha_{q_i}^M\}$ for each $q_i$ with Algorithm 1
5:     $\theta \leftarrow \theta + \frac{\eta}{MN} \sum\limits_{i=1}^{N} \sum\limits_{j=1}^{M} [r(a_i, \text{LLM}(q_i, \alpha_{q_i}^j)) \nabla_\theta \log P_\theta(\alpha_{q_i}^j | q_i)]$
6: **end while**
7: **return** $\theta$

---

### A.5 DESIGN MOTIVATION SUMMARY

In this section, we summarize the design motivation of AutoMR by first explaining the advantage of DAG-based meta reasoning skeleton over previous manually designed skeleton structure and then explaining the motivation of search algorithm for the DAG-based skeleton.

**Advantage of DAG-based skeleton: Capturing intricate logical dependencies among reasoning steps.** AutoMR adopts a DAG-based meta-reasoning skeleton. Unlike sequential, parallel, or tree-structured skeletons used in prior work, a DAG can subsume all of them, capture intricate logical dependency while avoiding inherent limitations of manually designed skeletons. Sequential skeletons force each step to attend to all previous steps, which can introduce hallucinations and error accumulation (Muennighoff et al., 2025; Sui et al., 2025). Parallel or tree structures isolate branches, preventing cross-branch information flow and leading to redundant reasoning steps that do not contribute to the final answer (Brown et al., 2024). A DAG naturally models logical dependencies via edges and can merge reasoning branches when appropriate, addressing these issues.

**Design motivation of search algorithm: Aligning with step-by-step reasoning process.** Given the advantages of the DAG structure, Section 3.2 focuses on searching for such a DAG. Common DAG search methods fix the number of nodes and optimizing all edge probabilities such as DARTS (Liu et al., 2019) for neural architecture search or GPTSwarm (Zhuge et al., 2024) for multi-agent workflow. They are unsuitable for LLM reasoning, because reasoning unfolds step by step rather than determination connections for predetermined reasoning steps. Moreover, meta-reasoning strategies must adapt to the evolving base-reasoning context. Therefore, Algorithm 1 incrementally expands the DAG node by node, without fixing its size, interleaving meta-reasoning with the step-by-step base-reasoning process

## B THEORETICAL ANALYSIS

### B.1 PROOF OF PROPOSITION 1

*Proof.* We prove each case by construction.

**Sequential.** A sequential structure is defined as an ordered set of noes $\mathcal{V} = \{v_1, \ldots, v_k\}$ with edges

$$\mathcal{E} = \{(i, i+1) \mid 1 \leq i \leq k-1\},$$

and $\tau((i, i+1) \in \mathcal{S}$ for each $i$. Clearly, $v_1$ is the unique source ($\deg^-(v_1) = 0$ and $\deg^-(v) = 1$ for all $v \neq v_1$), and $G$ is acyclic since edges only connect $v_i \to v_{i+1}$. Hence $(\mathcal{V}, \mathcal{E}, \mathcal{S}, \tau)$ is a single-source edge-heterogeneous DAG.

**Tree.** A tree is a rooted directed graph $G = (\mathcal{V}, \mathcal{E}, \mathcal{S}, \tau)$ such that:

$$\exists! \, r \in \mathcal{V} \text{ with } \deg^-(r) = 0, \quad \forall v \in \mathcal{V} \setminus \{r\}, \, \deg^-(v) = 1.$$

By definition, a rooted tree has no directed cycles and admits a unique source $r$. Since $\tau : \mathcal{E} \to \mathcal{S}$ can assign arbitrary heterogeneous edge types, $(\mathcal{V}, \mathcal{E}, \mathcal{S}, \tau)$ is a single-source edge-heterogeneous DAG.

**Parallel.** A parallel structure is defined by a common entry node $s$ and a family of disjoint branches

$$\mathcal{B} = \{B_1, \ldots, B_m\}, \quad B_i = (\mathcal{V}_i, \mathcal{E}_i, \mathcal{S}, \tau|_{\mathcal{E}_i}),$$

where $s \in \mathcal{V}$ and for each $i$ we have $(s, u) \in \mathcal{E}$ with $u \in \mathcal{V}_i$ the root of branch $B_i$. Thus the overall structure is

$$\mathcal{V} = \{s\} \cup \bigcup_{i=1}^{m} \mathcal{V}_i, \quad \mathcal{E} = \bigcup_{i=1}^{m} \left( \{(s, u_i)\} \cup \mathcal{E}_i \right).$$

This is precisely a rooted tree with root $s$ and subtrees $B_i$ attached as children. Therefore, a parallel structure is a *special case* of a tree, and hence also a single-source edge-heterogeneous DAG.

Since sequential, tree, and parallel (as a special case of tree) all admit representations $(\mathcal{V}, \mathcal{E}, \mathcal{S}, \tau)$ that satisfy (i) unique source, (ii) acyclicity, and (iii) heterogeneous edge labels, they are all contained in the class of single-source edge-heterogeneous DAGs. $\qquad\square$

### B.2 PROOF OF PROPOSITION 2

We first prove that Algorithm 1 can cover any skeleton $\alpha \in \mathcal{A}$ and then analyze the time complexity.

*Proof.* Since $\alpha$ is acyclic, by a standard result there exists a topological ordering of its vertices. That is, there exists a permutation $\pi = (n_1, n_2, \ldots, n_{|\mathcal{V}|})$ of $\mathcal{V}$ such that for every edge $(u \to w) \in \mathcal{E}$ we have $u$ appears earlier than $w$ in $\pi$.

Use this topological order $\pi$ as the insertion order in the append-only construction: add nodes in order $n_1, n_2, \ldots, n_{|}\mathcal{V}|$. When adding $n_t$, consider all previously added nodes $\{n_1, \ldots, n_{t-1}\}$. Because $\pi$ is a topological order, every edge in $\mathcal{E}$ that is incident to $n_t$ from earlier nodes is of the form $n_i \to n_t$ with $i < t$; there are no edges from $n_t$ back to any already-added node. Therefore, by choosing exactly those forward edges $\{(n_i \to v_t) \in \mathcal{E} \mid i < t\}$ at step $t$, we add precisely the edges of $\alpha$ that end at $n_t$.

Applying this procedure for $t = 1, \ldots, n$ adds all and only the edges of $\alpha$. Hence the append-only construction, with insertion order equal to any topological order of $\alpha$ and with edge choices equal to the edges of $\alpha$, reproduces $\alpha$ exactly. $\qquad\square$

Besides invoking the LLM to generate textual reasoning content, Algorithm 1 requires at most $O(|\mathcal{V}|^2)$ sampling process for reasoning steps count $|\mathcal{V}|$ with two layers of "for" loop, where each sampling process corresponds to a single MLP call.

Let $\mathcal{B}$ denote token budget of the generated reasoning content. Since Algorithm 1 introduces no additional LLM calls as analyzed in Section 3.2, the time complexity of LLM invocation remains $O(\mathcal{B}^2)$.

In practice, the reasoning step count $|\mathcal{V}|$ is roughly proportional to $\mathcal{B}$, but typically $|\mathcal{V}| \ll \mathcal{B}$, as each reasoning step consists of many tokens.

Furthermore, the computational cost of MLP inference is negligible compared with the layered blocks of the LLM. Therefore, AutoMR introduces only minimal additional computational overhead relative to naive LLM reasoning.

## C EXPERIMENT DETAILS

### C.1 BASELINE IMPLEMENTATION

The system prompt and answer extraction code for math Q&A problem is referred to a open-source repository **openr** [1]. The system prompt and answer extraction code for general multiple-choice problem is referred to the original MMLU-Pro repository [2].

For all baselines, we implement with Qwen and LLaMA as base model rather than the LLM used in their original paper for fair comparison.

- **MRP.** MRP dose not have open-source code, but provides prompt in original paper. We follow the paper to implement MRP.

- **Meta-Reasoner.** Meta-Reasoner dose not have open-source code, but provides prompt, pseudo code and detailed description in original paper. We follow the paper to implement Meta-Reasoner.

- **rStar.** We implement rStar with it open-source code [3].

- **MaAS.** We implement MaAS with it open-source code [4].

- **RS.** Referring to previous works (Bergstra & Bengio, 2012; Liu et al., 2019), we sample 48 architectures from search space randomly. Then we validate these architectures on training set to select the one with highest accuracy. With the selected architecture, we report its accuracy on test set.

- **QI.** Referring to previous works (Liu et al., 2019; Zhuge et al., 2024), we do not use an MLP which takes reasoning context as input and output meta strategy distribution, but model the strategy distribution of each edge in search space without condition. We optimize the distribution with the same estimation of policy gradient with REINFORCE as in Equation 4. For all queries in test set, we sample only one skeleton to process all of them.

- **CA.** Referring to previous works (Cheng et al., 2020; Zhang et al., 2025a), we use an MLP which takes semantic embedding of queries and meta reasoning strategies existing in skeleton as input to sample strategy for edges, rather than based on base reasoning context. For each query in test set, we sample a complete skeleton before inference and then reason for the query guided by the complete skeleton.

### C.2 DATASETS DETAILS

For training set, we use MATH [5] training split composed of 5053 query-answer pairs and MMLU-Pro [6] training split composed of 70 query-answer pairs. For testing set, we use GSM8K [7], MATH-500 [8], AMC [9], Olympiad [10] and four subset (Science, Humanities, Social and Other) of MMLU-Pro. We summarize the statistics of dataset in Tabel 4.

### C.3 ADDITIONAL RESULTS

#### C.3.1 ADDITIONAL RESULTS ON LARGER LLM

Besides Qwen2.5-3B-Inst and LLaMA3.2-3B-Inst used in Section 4.1 with token budget 1024, we conduct additional experiments to evaluate AutoMR based on larger long-cot LLM, including Qwen3-8B and Qwen3-14B (both enabling thinking mode, refer to Qwen3 homepage [11] for more

---

[1] https://github.com/openreasoner/openr
[2] https://github.com/TIGER-AI-Lab/MMLU-Pro
[3] https://github.com/zhentingqi/rStar
[4] https://github.com/bingreeky/MaAS
[5] https://github.com/hendrycks/math
[6] https://github.com/TIGER-AI-Lab/MMLU-Pro
[7] https://github.com/openai/grade-school-math
[8] https://huggingface.co/datasets/HuggingFaceH4/MATH-500
[9] https://huggingface.co/datasets/AI-MO/aimo-validation-amc
[10] https://github.com/OpenBMB/OlympiadBench
[11] https://qwenlm.github.io/blog/qwen3/

Table 4: Dataset Statistics.

| Domain | # Train | Dataset | # Test | Description |
|---|---|---|---|---|
| Math Q&A | 5053 | GSM8K | 1319 | Grade school math. |
| | | MATH-500 | 500 | High school math. |
| | | AMC | 83 | High school competition math. |
| | | Olympiad | 674 | Olympiad-level math competition. |
| General Multi-Choice | 70 | Science | 5345 | Physic, chemistry, biology, etc. |
| | | Humanities | 1981 | Philosophy, history and law. |
| | | Social | 2431 | psychology, business and economics. |
| | | Other | 924 | Other topics |

Table 5: Additional results on Qwen3-8B and Qwen3-14B.

| Method | MATH-500 | | Olympiad | |
|---|---|---|---|---|
| | Qwen3-8B | Qwen3-14B | Qwen3-8B | Qwen3-14B |
| Base (S) | 80.8 | 84.2 | 47.0 | 49.0 |
| Meta-Reasoner(S) | 81.6 | 85.6 | 49.9 | 50.4 |
| rStar (T) | 83.4 | 84.8 | 48.9 | 50.9 |
| AutoMR (G) | $86.6_{\uparrow 5.8}$ | $87.2_{\uparrow 3.0}$ | $53.9_{\uparrow 6.9}$ | $54.5_{\uparrow 5.5}$ |

details about thinking mode) with larger token budget 8192. We evaluate AutoMR and two meta reasoning methods that performs well in Table 1 on two datasets (MATH500 and Olympiad).

The results are showed in Table 5, where **Base** means the thinking mode of Qwen3. These results show that AutoMR still achieves performance gains when scaling to larger LLMs with larger token budget, compared with naive LLM thinking and previous meta reasoning methods, demonstrating the scaling potential of AutoMR.

### C.3.2 ADDITIONAL RESULTS ON EXTRA DATASETS

Besides the math datasets and multiple-choice datasets in Section 4.1 we conduct additional experiments based on Qwen3-8B on three complex reasoning, including Game of 24, BIG-Bench Hard (BBH) and Python Programming Puzzles (P3). We evaluate AutoMR and two meta reasoning methods that performs well in Table 1 on these datasets.

The results are showed in Table 6, where **Base** means the thinking mode of Qwen3-8B. The results show that AutoMR still achieves better overall performance on these three datasets. Specifically, the advantage is most significant on P3, followed by BBH. The advantage is not large on Game of 24. We claim that such difference is because these datasets requires completely different reasoning patterns.

On Game of 24, we find that all methods demonstrate similar strategies across different queries, namely exploring all combinations of four given numbers and +, -, *, /, and checking these combinations one by one. AutoMR also searches out such meta reasoning skeleton. So performance of these methods is similar on Game of 24.

However, queries from P3 requires much more complex meta reasoning strategies, including exploring new solutions or reflecting the proposed solution. This makes DAG-based meta reasoning skeleton proposed by AutoMR much more effective than the baselines.

### C.3.3 ADDITIONAL RESULTS OF OTHER BASELINES.

Besides the baselines in Section 4.1, we also compare AutoMR with other two relevant works, i.e. Buffer-of-Thought (BoT) (Yang et al., 2024) and Graph-of-Thought (GoT) (Besta et al., 2024).

BoT utilizes a meta-buffer to store and retrieve high-level thought-templates distilled from various problem-solving processes. For BoT, we evaluate based on Qwen3-8B, on five datasets, including

Table 6: Additional results on Game of 24, BBH and P3, based on Qwen3-8B.

| Method | Game of 24 | BBH | P3 |
|---|---|---|---|
| Base (S) | 73.0 | 80.2 | 75.0 |
| Meta-Reasoner(S) | 75.0 | 83.9 | 76.8 |
| rStar (T) | 77.0 | 80.6 | 79.6 |
| AutoMR (G) | $\mathbf{79.0}_{\uparrow 6.0}$ | $\mathbf{85.6}_{\uparrow 5.4}$ | $\mathbf{84.4}_{\uparrow 9.4}$ |

Table 7: Additional results BoT based on Qwen3-8B.

| Method | MATH500 | Olympiad | Game of 24 | BBH | P3 |
|---|---|---|---|---|---|
| Base (S) | 80.8 | 47.0 | 73.0 | 80.2 | 75.0 |
| BoT-EmptyInit | $81.0_{\uparrow 0.2}$ | $47.4_{\uparrow 0.4}$ | $76.0_{\uparrow 3.0}$ | $\mathbf{85.7}_{\uparrow 5.5}$ | $80.6_{\uparrow 5.6}$ |
| BoT | $83.6_{\uparrow 2.8}$ | $52.1_{\uparrow 5.1}$ | $\mathbf{94.0}_{\uparrow 21.0}$ | $\mathbf{85.7}_{\uparrow 5.5}$ | $80.6_{\uparrow 5.6}$ |
| AutoMR | $\mathbf{86.6}_{\uparrow 5.8}$ | $\mathbf{53.9}_{\uparrow 6.9}$ | $79.0_{\uparrow 6.0}$ | $85.6_{\uparrow 5.4}$ | $\mathbf{84.4}_{\uparrow 9.4}$ |

MATH-500, Olympiad, Game of 24, BBH and P3. Considering that the open-source code of BoT provides some templates to initialize meta buffer as well as update it dynamically by prompting LLM summarize new templates, for fair comparison, we also design a variant named **BoT-EmptyInit** that initializes meta buffer without provided templates but reserve the dynamic update process. Since the open-source code of BoT dose not provide templates for P3 and most subsets of BBH, we implement BoT as equivalent to BoT-EmptyInit on these datasets.

The results are showed in Table 7. According to Table 7, AutoMR performs better than BoT on MATH500, Olympiad and PPP. The performance is similar on BBH and BoT outperforms AutoMR on Game of 24.

This is because that BoT retrieves thought template from meta buffer and reasons according to that template. Thought template is highly effective when it matches given query. However, thought template is not robust across different queries. Even though there is only a little modification to a query, this may lead to template ineffective. On Game of 24, the combination of four numbers and +, -, *, / is finite, so there is a general template by checking all combinations exhaustively. Moreover, BoT provides manually designed python code to implement such exhaustive process without relying on token budget. Therefore, BoT performs much better on Game of 24.

However, on math problem (MATH500, Olympiad) and and code task (P3), these queries is quite diverse, making the templates designed manually or summarized by LLM hard to generalize to different queries. Therefore, AutoMR performs better on these three datasets.

GoT also uses graph to represent reasoning process to improve reasoning performance. However, it is quite different from AutoMR. We summarize the main difference as follows:

- **Graph of AutoMR and GoT has different meanings and granularities**. GoT's graph has different meanings from AutoMR. For example, GoT follows the paradigm that decomposes the task into several subtasks and solve them respectively. Each node in its graph represents a solution to a subtask, rather than a reasoning step as in AutoMR. So AutoMR is more fine-grained. Moreover, the edge in GoT only represents dependency, while the edge also represents meta reasoning strategy in AutoMR.

- **GoT requires user to design graph structure and configure it manually**. GoT in fact provides an interface for the user to design a static graph structure and related configuration manually but not build a thought graph itself automatically.

- **GoT is not suitable to tasks in our experiments**. The graph indicating how to reason in GoT is static, this makes GoT only applicable to tasks where queries can be solved by fixed reasoning patterns (such as merge-based sorting for array with fixed length). This makes it struggle in tasks requiring diverse reasoning patterns used in our experiment (such as math).

Table 8: Additional results GoT based on Qwen3-8B.

| Method | MATH500 | Olympiad |
|--------|---------|----------|
| Base (S) | 80.8 | 47.0 |
| GoT | $78.6_{\downarrow 2.2}$ | $45.7_{\downarrow 1.3}$ |
| AutoMR | $\mathbf{86.6}_{\uparrow 5.8}$ | $\mathbf{53.9}_{\uparrow 6.9}$ |

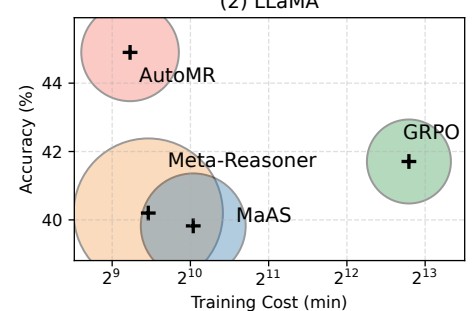

Figure 6: The training and inference cost and performance of AutoMR and baselines.

Despite this, we implement a thought graph manually based on interface provided by GoT and evaluate it based on Qwen3-8B on MATH500 and Olympiad. The results are showed in Table 8. The results show that GoT cannot work well for the tasks of interest in our experiment.

### C.3.4 DISCUSSION ABOUT COMPARISON WITH LORA

AutoMR optimizes only an MLP to search for meta reasoning skeleton, rather than training LLM parameters directly. This is relevant to parameter efficient fine-tuning (PEFT) technique, which can be represented by LoRA (Hu et al., 2022). In this section, we discuss the niche of AutoMR compared with LoRA as follows:

- LoRA is a PEFT technique which can be widely used for various downstream tasks, by tuning LLM to this task efficiently. However, it dose not consider the characteristics of specific task.
- AutoMR is designed tailored to LLM reasoning task, considering the characteristics of reasoning task. Thus it performs better than LoRA when applied to LLM reasoning task.

Concretely, the advantage of AutoMR over LoRA on LLM reasoning task can be summarized as **comparable or even higher performance**, **higher training efficiency** and **higher data efficiency**. We demonstrate these advantages empirically.

In Section 4.3 about training and inference efficiency, we report the accuracy on MATH500 datasets in Figure 4. In this section, we report the average accuracy of AutoMR and other baselines including GRPO with LoRA, as well as training and inference cost in Figure 6. According to Figure 6, AutoMR requires much less training time than GRPO with LoRA, demonstrating the training efficiency of AutoMR. AutoMR achieves comparable accuracy based on Qwen, but much higher accuracy based on LLaMA, than GRPO with LoRA.

To validate the data efficiency of AutoMR,we conduct additional experiments by sampling 100 training samples randomly from the training set of math Q&A problem. We use the 100 samples to train AutoMR and train the LLM directly with GRPO by LoRA. We implement this based on Qwen and evaluate them on MATH500. We summarize the results in Table 9. The results show that AutoMR achieves higher performance than training the LLM directly with LoRA, demonstrating the data-efficiency of AutoMR. We illustrate the data-efficiency of AutoMR as follows.

- **Meta reasoning methods reduce data and computation requirement by heuristic design**. Current methods use RL to enable LLM to reason. These methods observe that meta reasoning behaviors is important in improving LLM reasoning ability (DeepSeek-AI, 2025). They emerges without explicit constraint but usually late in RL training phase (Wang et al., 2025). Meta rea-

Table 9: Results of AutoMR and LoRA with 100 training samples.

| Method | MATH500 |
|--------|---------|
| CoT (S) | 61.6 |
| LoRA | $62.8_{\uparrow 1.2}$ |
| AutoMR | $\mathbf{64.8}_{\uparrow 3.2}$ |

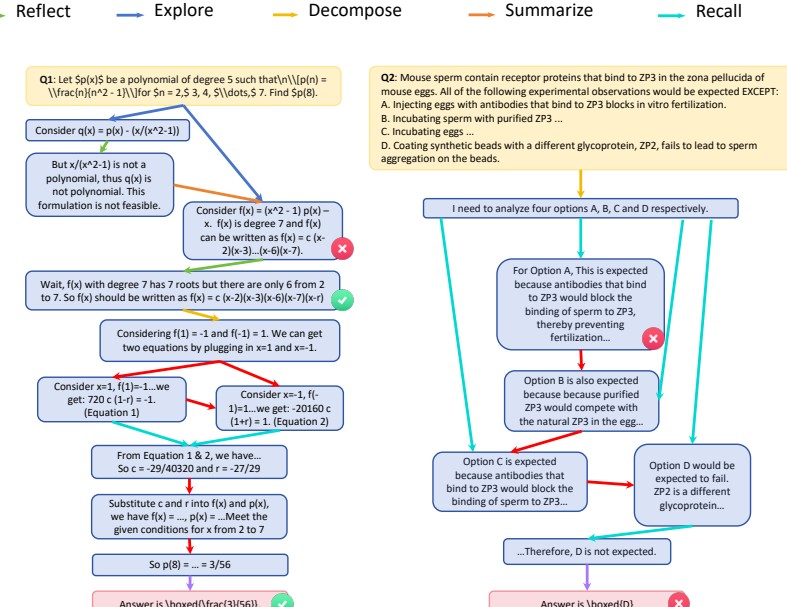

Figure 7: Searched skeletons for two queries. The left (from MATH500) is a successful case while right (from Science) is a failed case.

soning methods incorporate such behaviors designed heuristically into LLM reasoning process directly, rather than let them emerge spontaneously during RL training. Therefore, meta reasoning methods require relatively lower data and computation.

- **AutoMR further reduces the training difficulty by transforming generation into discrimination**. Rather than learn to generate meta-reasoning behaviors, the model only needs to select among predefined strategies. This discrimination task is substantially easier than generation, leading to lower data and model complexity.

### C.3.5  ADDITIONAL CASES

In Section 4.4, we show three cases to show how AutoMR improves LLM reasoning performance across queries from different disciplines and difficulties. In this section, we show additional cases (one successful case and one failed case) to show how AutoMR improves reasoning performance of naive CoT and the limitation of AutoMR we have observed.

As shown in Figure 7 left, step 3 (marked with a red cross) is a error step that is often reached by both AutoMR and naive CoT. In this step, "$f(x)$ is degree 7" while its factorization form "$f(x) = c(x-2)(x-3)...(x-6)(x-7)$" in fact has only 6 roots. However, in naive CoT, LLM will usually follow this error step for following reasoning process and reaches the error result in multiple runs. On the contrary, AutoMR samples "Reflection" strategy for edge (3, 4) as shown in Figure 7 and find out this error to fix it in step 4 (marked with a green tick). Finally AutoMR can reach the correct answer.

As shown in Figure 7 right, in the step for option A (marked with a red cross), AutoMR prompts LLM to recall related knowledge about the option. LLM successfully recall that "antibodies that bind to ZP3 would block the binding of sperm to ZP3". However, it overlooks that the zona pellucida

is on the surface of mouse eggs. Injected antibodies will not bind to ZP3 on the surface. Finally, this error step propagates to the final answer and makes it wrong. This case shows that similar to many current LLM reasoning methods including reinforcement learning, AutoMR can improve LLM reasoning performance but it may not be able to break through the ability boundary of base LLM.

## D   USE OF LLMS

We use LLMs only to polish writing grammatically. We review and revise all content generated by LLMs to ensure accuracy.

