# OpenReview forum: "Searching Meta Reasoning Skeleton to Guide LLM Reasoning"
_ICLR.cc/2026/Conference — Submitted to ICLR 2026_

### Official Review · Reviewer_CbsY · 2025-10-14

**Soundness:** 3
**Presentation:** 2
**Contribution:** 3
**Rating:** 2
**Confidence:** 3

**Summary:**

AutoMR reframes “meta reasoning” for LLMs as searching a query-aware **reasoning skeleton** represented by a single-source, edge-heterogeneous DAG that can encode sequential, parallel, tree, and more intricate dependencies among steps. It defines a strategy set (e.g., Next, Reflect, Explore, Decompose, Summarize, Recall, Answer) and introduces a **dynamic skeleton sampling** algorithm that interleaves with inference: for each step it selects incoming strategy-typed edges conditioned on the evolving reasoning context, generating content only when needed. A lightweight MLP guides edge selection, and a policy over skeletons is optimized with REINFORCE; this adds minimal overhead compared to vanilla reasoning while adapting structure per query. Across math QA (GSM8K, MATH-500, AMC, Olympiad) and general multiple-choice (MMLU-Pro) with LLaMA-3B and Qwen-3B backbones under the same token budgets, AutoMR consistently outperforms CoT, MRP, Meta-Reasoner, rStar, and an agent-workflow NAS baseline, and scales compute more efficiently, highlighting the advantage of DAG-based, instance-specific meta-reasoning.

**Strengths:**

1. The authors’ motivation is well founded, where for different problems, the meta-reasoning framework should be dynamic rather than static.
2. The proposed method is highly effective, achieving substantial performance gains across settings compared with prior approaches.
3. While improving accuracy, the method also demonstrates better efficiency than previous work.

**Weaknesses:**

1. Experiments are limited to short-CoT and small-scale models, which constrains the validation of the approach. Evaluating on long-CoT models (e.g., DeepSeek-R1-Distilled, Qwen3) and larger models (e.g., 8B, 14B) would more convincingly substantiate the method’s effectiveness.
2. Although many prior works also rely on training for meta reasoning, in many practical scenarios it is difficult to obtain sufficient training data and compute. Even when such resources are available, one could instead train a LoRA to boost performance, which makes the niche for this work somewhat awkward.
3. The method bears similarities to Graph-of-Thought [1]. Although the granularity differs, both enhance performance by structuring the reasoning process as a graph.
4. The writing is somewhat disorganized: for example, Figure 1 is not placed at the top, and Figure 5 appears to be a Table.

[1] Graph of Thoughts: Solving Elaborate Problems with Large Language Models. Besta et al., AAAI 2024.

**Questions:**

1. How did you determine the set of strategies used in Table 2? Why not include more or fewer strategies?
2. What are the MLP hyperparameters in your experiments?
3. Why does Figure 4 report results only on MATH-500 rather than an average across all datasets?
4. It is surprising that on MMLU-Pro such significant gains are achieved with only 70 training examples. Did you attempt to train the MLP on MATH-500 with fewer than 100 training samples as well?

---

> ### Author Response · Authors · 2025-11-25
>
> We appreciate the reviewer’s time and valuable observations.
>
> > **W1.** Experiments are limited to short-CoT and small-scale models, which constrains the validation of the approach. Evaluating on long-CoT models (e.g., DeepSeek-R1-Distilled, Qwen3) and larger models (e.g., 8B, 14B) would more convincingly substantiate the method’s effectiveness.
>
> **A1.** Thank you for your suggestion about validating AutoMR on larger long-CoT LLMs.
>
> We follow this suggestion to conduct additional experiments with two larger long-CoT LLMs (Qwen3-8B and Qwen3-14B), on two datasets (MATH500 and Olympiad). We compare AutoMR with Qwen3-8B thinking mode and the meta reasoning baselines that performs well according to Table1. We summarize the results as follows:
>
> ---
>
> | Qwen3-8B-thinking | MATH500  | Olympiad |
> | - | -| - |
> | Base | 80.8     | 47.0     |
> | rStar             | 83.4     | 48.9     |
> | Meta Reasoner     | 81.6     | 49.9     |
> | **AutoMR**        | **86.6** | **53.9** |
>
> ---
>
> | Qwen3-14B-thinking | MATH500  | Olympiad |
> | -| -| -|
> | Base| 84.2     | 49.0 |
> | rStar| 84.8     | 50.9  |
> | Meta Reasoner| 85.6     | 50.4     |
> | **AutoMR**  | **87.2** | **54.5** |
> ---
>
> According to the results, **AutoMR** still outperforms the baselines even when scaling to larger long-CoT LLM (i.e. 8B and 14B), demonstrating the scaling potential of AutoMR.
>
> We include these results and analysis in Appendix C.3.1 of the updated manuscript.
>
>
> > **W2.** Although many prior works also rely on training for meta reasoning, in many practical scenarios it is difficult to obtain sufficient training data and compute. Even when such resources are available, one could instead train a LoRA to boost performance, which makes the niche for this work somewhat awkward.
>
> **A2.** Our response has two layers. First, we clarify the practical value of meta-reasoning relative to directly training LoRA; second, we detail AutoMR’s specific advantages within the meta-reasoning paradigm.
>
> (1) Why meta reasoning is useful (vs. only training LoRA).
>
> - **Implement “how to think” explicitly, thus data/compute friendly.** Prior works[1, 2] show meta reasoning behaviors like exploration/verification/reflection are pivotal for reasoning and often emerge only late in RL without explicit supervision. Meta reasoning injects these behaviors procedurally at inference time or via a light controller, rather than hoping they emerge from heavy fine-tuning, thus avoiding large-scale parameter updates and long training runs compared with training LLM directly.
>
> - **Complementary to LoRA.** Meta reasoning and LoRA are not mutually exclusive: a controller can sit atop a LoRA-adapted base; our question here is whether meta reasoning alone has a clear niche—answer: yes, it brings procedural control, better test-time adaptability, and lower training burden.
>
>
> (2) Why AutoMR is a strong meta reasoning method.
>
> - **Captures complex dependencies.** Unlike fixed, hand-written skeletons, AutoMR searches single-source, edge-heterogeneous DAGs, capturing non-linear dependencies among steps (branches, verification, backtracking).
>
> - **Query-specific at test time.** AutoMR interleaves sampling with the evolving base-level state, yielding a per-query skeleton instead of a one-size-fits-all template.
>
> - **No extra supervision.** As formalized in Section 3.2 (Definition 1), AutoMR needs only final-answer rewards—no behavior labels or reason-trace annotations—on par with standard RL/LoRA pipelines.
>
> - **Data-efficiency in low-data regimes.** Our additional experiments (see A8 for Q4) show that learning a small policy over behaviors can outperform LoRA when data is scarce, because AutoMR reduces the learning burden from model parameters to a compact meta-policy.
>
> - **Compute-efficiency.** AutoMR does not fine-tune the base LLM and adds no extra LLM calls beyond standard reasoning. Proposition 2 (line 264) shows minimal theoretical overhead (details in Appendix B.2). Empirically, Figure 4 (Section 4.3) shows lower training cost than LoRA at comparable or better accuracy.
>
> In summary, meta reasoning is valuable because it makes the reasoning process controllable and data/compute-efficient; within that space, AutoMR advances the state of the art by searching query-specific DAG skeletons that capture complex dependencies with minimal overhead and strong low-data performance.
>
> [1] Guo D, Yang D, Zhang H, et al. Deepseek-r1: Incentivizing reasoning capability in llms via reinforcement learning. 2025
>
> [2] Haozhe Wang, et al. Emergent Hierarchical Reasoning in LLMs through Reinforcement Learning. 2025

---

> ### Author Response · Authors · 2025-11-25
>
> (Cont.)
> > **W3.** The method bears similarities to Graph-of-Thought [1]. Although the granularity differs, both enhance performance by structuring the reasoning process as a graph.
>
> **A3.** Graph-of-Thought (GoT) indeed uses graph to represent reasoning process. However, the idea, technique and applicable tasks are distinctly different besides granularity. We explain these differences as follows.
>
> 1. **Graph of AutoMR and GoT has different meanings and granularities**. GoT follows the paradigm that decomposes task into sub-tasks  and solve these sub-tasks respectively. So the nodes in GoT' graph represents solutions to sub-tasks. However, the nodes in AutoMR's graph represents reasoning steps. As you have mentioned, the graph in AutoMR is more fine-grained than that in GoT.
>
> 2. **GoT requires user to design graph structure and configure it manually**. GoT can not build a graph to organize LLM reasoning process automatically. In contrast, it in fact provides an interface for the user to design the graph manually. The user need to design the graph structure and set the subtasks that need to be solved by each node. However, AutoMR dose not need the human design.
>
> 3. **GoT is not suitable to tasks in our experiments**. The graph indicating how to reason in GoT is static if without user's manual intervention, this makes GoT only suitable to tasks where queries can be solved by fixed reasoning patterns (such as merge-based sorting for array with fixed length), and struggle in tasks requiring diverse reasoning patterns used in our experiment (such as math). GoT allows removing some nodes to change graph structure during reasoning process, but this requires user's manual intervention.
>
> Despite the substantial differences from our approach, we still managed to compare against GoT as suggested.
> We implement a thought graph manually based on GoT and evaluate it with Qwen3-8B on MATH500 and Olympiad. We summarize the results as follows.
>
> ---
>
> | Qwen3-8B-thinking | MATH500 | Olympiad |
> | ----------------- | ------- | -------- |
> | Base              | 80.8    | 47.0     |
> | **GoT**           | 78.6    | 45.7     |
> | **AutoMR**        | 86.6    | 53.9     |
>
> ---
>
> The results show that GoT cannot work well for the tasks of interest in our experiment. We include these results and analysis in the updated Appendix C.3.3.
>
>
>
>
> > **W4.** The writing is somewhat disorganized: for example, Figure 1 is not placed at the top, and Figure 5 appears to be a Table.
>
> **A4.** Thank you for your suggestion about writing, we fix these errors in updated version.
>
>
>
> > **Q1.** How did you determine the set of strategies used in Table 2? Why not include more or fewer strategies?
>
> **A5.** We refer to recent works (such as [1~3]) about LLM reasoning or LLM cognition and summarize the meta reasoning strategies that widely appear in general. According to researches about cognition science [4], the meta reasoning strategies is quite broad. Thus it is impossible for a finite set to cover all cognition behaviors. But fining out a set suitable for given task is feasible. Empirically, including the strategies in Table 2 **(Table 3 in the updated version)** achieves good performance in  our experiments. We do not include some other strategies mentioned in reviewed works because they are not widely used in general or they may reduce the performance in our experiments.
>
> Moreover, the set in AutoMR is quite extensible and flexible. When a new meta reasoning behavior is found effective for given task, it can be added to the set easily. On the contrary, a strategy effective in previous task can be removed if not effective in new tasks. It is opento adjust the set according to task characteristics.
>
> [1] Gandhi K, Chakravarthy A, Singh A, et al. Cognitive behaviors that enable self-improving reasoners, or, four habits of highly effective stars. COLM 2025.
>
> [2] Chen Q, Qin L, Liu J, et al. Towards reasoning era: A survey of long chain-of-thought for reasoning large language models. 2025.
>
> [3] Luo Y, Song Y, Zhang X, et al. Deconstructing long chain-of-thought: A structured reasoning optimization framework for long cot distillation. 2025.
>
> [4] Flavell, J. H. Metacognition and cognitive monitoring: A new area of cognitive-developmental inquiry. 1979.
>
>
>
> > **Q2.** What are the MLP hyperparameters in your experiments?
>
> **A6.** The hyperparameters of our MLP in experiments depend on the LLM we use. For an LLM with hidden size $d$, the input dim of our MLP is $3*d$. Then we usually use a three-layer MLP that reduce the dim to $2 \times d$, $d$, $d / 2$ and finally to the number of meta reasoning behaviors. Our MLP model is in fact a classification head that outputs the probability of each meta reasoning strategy.
>
> We include this implementation details in updated Appendix A.4.

---

> ### Author Response · Authors · 2025-11-25
>
> > **Q3.** Why does Figure 4 report results only on MATH-500 rather than an average across all datasets?
>
> **A7.** In Figure 4, we report the accuracy on MATH500, as well as training cost and inference cost on all math problems. As described in Section 4.1 about experiment set up, take math Q&A task as an example, we use only one training set and train only one model on it. Then we evaluate the model on four different datasets, including MATH500, GSM8K, AMC and Olympiad. So the training cost in Figure 4 is not on MATH500, but on the training set shared by four test dataset. As for accuracy for MATH500 in Figure 4, it is the same as in Table 1 and the accuracy of other dataset is also listed in Table 1. The inference cost is the average across all test datasets.
>
> We add results with average accuracy in Appendix C.3.4 as in Figure 6.
>
>
>
> > **Q4.** It is surprising that on MMLU-Pro such significant gains are achieved with only 70 training examples. Did you attempt to train the MLP on MATH-500 with fewer than 100 training samples as well?
>
> **A8.** Thank you for your insight about the data efficiency of AutoMR.
>
> Following this insight, we conduct additional experiments by sampling 100 training samples randomly from the training set of math Q&A problem. We use the 100 samples to train **AutoMR** and train the LLM with GRPO by LoRA. We implement this based on  Qwen2.5-3B-Inst and evaluate them on MATH500. We summarize the results as follows.
>
> ---
>
> | Qwen2.5-3B-Inst | MATH500      |
> | --------------- | ------------ |
> | CoT             | 61.6         |
> | **LoRA-100**    | 62.8 (+1.2%) |
> | **AutoMR-100**  | 64.8 (+3.2%) |
>
> ---
>
> The results show that AutoMR achieves higher performance than training the LLM directly with LoRA, demonstrating the data-efficiency of AutoMR. We illustrate the data-efficiency of AutoMR as follows.
>
> * **Meta reasoning reduces data and computation requirement heuristically.** In response **A2** to **W2**, we illustrate that meta reasoning behaviors have been observed important in improving LLM reasoning ability. Meta reasoning methods incorporate such behaviors designed heuristically into LLM reasoning process directly, rather than let them emerge spontaneously during RL training. Therefore, meta reasoning methods require relatively lower data and computation.
>
> * **AutoMR further reduces the training difficulty by transforming generation into discrimination**. Rather than learning to generate meta-reasoning behaviors, the model only needs to select among predefined strategies. This discrimination task is substantially easier than generation, leading to lower data and model complexity.
> * **Searching skeleton on MMLU-Pro is easier than on math tasks.** The performance gains with 100 samples are smaller than on MMLU-Pro. This is because that the skeletons for general multiple-choice task have simpler structure. We find that many skeletons follow the paradigm that decomposing the query into dealing with each option one-by-one and then recalling knowledge for each option respectively. While some uncertain options may be summarized for final choice. This pattern makes it easier to train on this task.

---

> ### Comment · Reviewer_CbsY · 2025-11-25
>
> Thank you for the detailed clarifications and additional experiments provided in the rebuttal, which do alleviate some of my concerns regarding scalability and implementation details to a certain extent. However, after considering the response as a whole, I feel that some of my original core concerns remain not fully resolved. For example, with respect to systematic validation across a broader range of tasks and model scales, the new results still seem to focus mainly on a limited set of datasets and model families; compared with more common practical approaches such as directly applying or strengthening LoRA, the concrete advantages and applicability boundaries of the proposed method in real-world scenarios are still somewhat unclear to me. In addition, although the authors have further differentiated AutoMR from Graph-of-Thought and provided corresponding comparison experiments, at a conceptual level I still feel that the novelty lies more in the specific implementation and formalization than in a clear paradigm shift. Overall, I appreciate the efforts made in the rebuttal and agree that the work has potential; however, relative to my bar for acceptance, these changes are not sufficient to substantially alter my overall assessment, and I thus prefer to keep my original score.

---

> > ### Author Response · Authors · 2025-12-02
> >
> > Thank you for your timely comments.
> >
> > **Results on other datasets and LLM families.** We conduct additional experiments based on DeepSeek-R1-Distill-Llama-7B and Qwen3-8B on more datasets, including additional Game of 24, Big-Bench-Hard (BBH) and Python Programming Puzzles. We summarize the results as follows. The results show that AutoMR generalize well to other LLM families and reasoning tasks.
> >
> > ---
> >
> > | Qwen3-8B-thinking | MATH500  | Olympiad | Game of 24 | BBH      | P3       |
> > | ----------------- | -------- | -------- | ---------- | -------- | -------- |
> > | Base              | 80.8     | 47.0     | 73.0       | 80.2     | 75.0     |
> > | rStar             | 83.4     | 48.9     | 75.0       | 83.9     | 76.8     |
> > | Meta Reasoner     | 81.6     | 49.9     | 77.0       | 80.6     | 79.6     |
> > | **AutoMR**        | **86.6** | **53.9** | **79.0**   | **85.6** | **84.4** |
> >
> > ---
> >
> > | DeepSeek-R1-Distill-Llama-7B | MATH500 | Olympiad | Game of 24 | BBH  | P3   |
> > | ---------------------------- | ------- | -------- | ---------- | ---- | ---- |
> > | Base                         | 77.6    | 40.2     | 60.0       | 48.9 | 47.4 |
> > | rStar                        | 78.6    | 41.3     | 63.0       | 51.1 | 49.7 |
> > | Meta Reasoner                | 76.4    | 39.0     | 62.0       | 47.6 | 48.3 |
> > | **AutoMR**                   | **80.8**    | **43.3**    | **68.0**      | **54.2** | **51.7** |
> >
> > ---
> >
> > **Lora is a PEFT technique for LLM while AutoMR is tailored to LLM Reasoning Task.** Lora is a PEFT technique which can be widely used for various downstream tasks, by tuning LLM to this task efficiently. But it dose not consider the characteristics of specific task. AutoMR is designed tailored to LLM reasoning task, thus it achieves higher reasoning performance, higher data-efficiency and higher training-efficiency than Lora on LLM reasoning task. We have demonstrated these advantages with experiments (as shown in Figure 4 and A8). The two methods are orthogonal to each other.
> >
> > **The main paradigm gap between AutoMR and GoT lies not in using what structure to model reasoning process, but in that the reasoning process is regulated by manual configuration or explored by LLM itself automatically.** GoT just provides an graph-based interface for user to define how the LLM should reason manually. On the contrary, AutoMR lets LLM to generate a graph-based meta reasoning skeleton itself automatically. This gap also causes that GoT cannot work well on the reasoning tasks (e.g. mathematics) without fixed reasoning template in our work. In fact, except for not the same graph formulation (AutoMR is edge-heterogeneous to represent meta reasoning strategy and more fine-grained), the two methods have nothing similar to each other.

---

### Official Review · Reviewer_SgMk · 2025-10-19

**Soundness:** 3
**Presentation:** 3
**Contribution:** 2
**Rating:** 4
**Confidence:** 3

**Summary:**

This paper proposes AutoMR, a framework designed to search for meta-reasoning skeletons more effectively and efficiently. Experiments conducted across a wide range of tasks demonstrate the framework’s strong performance and provide a comprehensive analysis of its effectiveness.

**Strengths:**

- The presentation and writing are clear and generally easy to follow.

- The core idea and motivation are interesting and relevant.

- The proposed method appears highly effective.

- The experiments and analyses are comprehensive and well-executed.

**Weaknesses:**

- Design Motivation: The paper would benefit from a deeper explanation of the design motivation behind AutoMR. Many parts of the explanation are highly technical and provide limited intuition for understanding why the framework works. While the inclusion of formulas is appreciated, the methodology—especially Section 3.2.1—would be easier to follow if accompanied by more intuitive explanations or illustrative examples.

- Baselines: The paper should include comparisons with more recent and relevant reasoning models to strengthen its empirical claims.

- Analysis Depth: Additional analysis could enhance the interpretability and transparency of the framework, helping readers gain a deeper understanding of its internal mechanisms and behavior. (e.g., error analysis to understand the limitations)

I would be willing to raise my score if the authors can adequately address these weaknesses.

**Questions:**

The meta-reasoning behaviors summarized in the paper are derived from existing studies on LLM reasoning. Are these behaviors comprehensive enough? Have the authors considered incorporating other types of reasoning behaviors, perhaps inspired by cognitive or psychological perspectives?

---

> ### Author Response · Authors · 2025-11-25
>
> We thank the reviewer for the careful evaluation and insightful suggestions.
>
> > **W1.** Design Motivation
>
> **A1.** Thank you for your suggestion to make the methodology easier to follow.
>
> First, we explain the design motivation of methodology. Then, we use Figure 2 as an example to intuitively show the flow of methodology in Section 3.2.1.
>
> **Design Motivation**: We explain the design motivation by first explaining the advantage of DAG-based meta reasoning skeleton over previous manually designed skeleton structure and then explaining the motivation of search algorithm for the DAG-based skeleton in Section 3.2.1
>
> 1. **Advantage of DAG-based skeleton: Capturing intricate logical dependencies among reasoning steps**. AutoMR adopts a DAG-based meta-reasoning skeleton. Unlike sequential, parallel, or tree-structured skeletons used in prior work, a DAG subsumes all of them while avoiding their inherent limitations. Sequential skeletons force each step to attend to all previous steps, which can introduce hallucinations and error accumulation [1]. Parallel or tree structures isolate branches, preventing cross-branch information flow and leading to redundant reasoning steps that do not contribute to the final answer [2]. A DAG naturally models logical dependencies via edges and can merge reasoning branches when appropriate, addressing these issues.
>
> 2. **Design motivation of search algorithm in Section 3.2.1: Aligning with step-by-step reasoning process**. Given the advantages of the DAG structure, Section 3.2.1 focuses on searching for such a DAG. Common DAG search methods fix the number of nodes and optimizing all edge probabilities such as DARTS [3] for neural architecture search or GPTSwarm [4] for multi-agent workflow. They are unsuitable for LLM reasoning, because reasoning unfolds step by step and meta-reasoning strategies must adapt to the evolving base-reasoning context (as noted around line 211). Therefore, Algorithm 1 incrementally expands the DAG node by node, without fixing its size, interleaving meta-reasoning with the step-by-step base-reasoning process
>
> **Figure 2 as An Example**: We will use Figure 2 Top as an example to illustrate Section 3.2.1 intuitively. In Figure 2, we show the process of sampling a skeleton with 3 nodes (i.e. 0, 1 and 2. Node 3 is not added to skeleton finally.)
>
> 1. **Initially**, the skeleton is initialized with only Node 0. Node 0 represents the given query. Then AutoMR expand the skeleton node by node through sampling incoming edges for them, until token budget is reached or completion condition (all incoming edges for one node are *zero* type) is satisfied.
> 2. **Adding Node 1**, shown in Figure 2 Top (1).
>
>    (a). **Step 1: Determine incoming edges for Node 1 (Line 3~5 in Algorithm 1)**. AutoMR checks each existing node (but right now only Node 0) to decide if edge from the existing node to Node 1 should exist and which strategy to use (line 4 in Algorithm 1). In this example, the blue edge is sampled for edge (0, 1).
>
>    (b). **Step 2: Check completion (Line 6~8 in Algorithm 1)**. The sampled strategy is not a *zero* strategy, thus not complete.
>
>    (c). **Step 3: Generate content for Node 1 (Line 9 in Algorithm 1)**. AutoMR generates content for node 1 based on reasoning context and sampled strategy, and add it to the skeleton.
>
> 3. **Adding Node 2**, shown in Figure 2 Top (2).
>
>    (a). **Step 1: Determine incoming edges for Node 2 (Line 3~5 in Algorithm 1)**. The process is the same as adding Node 1. The difference is that there are two existing nodes (0 and 1). For edge (0, 2) and (1, 2), we sample strategies for them sequentially. In Figure 2, red is sampled for (1, 2) and blue is sampled for (0, 2).
>
>    (b). **Step 2: Check completion (Line 6~8 in Algorithm 1)**. There is no gray edge (*zero* strategy) sampled, thus not complete.
>
>    (c). **Step 3: Generate content for Node 2 (Line 9 in Algorithm 1)**. AutoMR generates content for node 2 and add it to the skeleton.
>
> 4. **Adding Node 3**, shown in Figure 2 Top (3).
>
>    (a). **Step 1: Determine incoming edges for Node 3 (Line 3\~5 in Algorithm 1)**. We repeat the process for Node 3 by sampling strategy for edge (2, 3), (1, 3) and (0, 3) sequentially (line 3\~5 in Algorithm). However, all sampled strategies are *zero* (i.e. gray edge in Figure 2).
>
>    (b). **Step 2: Check completion (Line 6~8 in Algorithm 1)**. Since all strategies are *zero*, the skeleton is complete.
>
> 5. **Generate Final Answer (Line 11 in Algorithm 1)**. AutoMR generates final answer according to reasoning process.
>
> [1] Muennighoff N, et al. s1: Simple test-time scaling, EMNLP 2025.
>
> [2] Brown, et al., and Mirhoseini, A. Large language monkeys: Scaling inference compute with repeated sampling, 2024.
>
> [3] Liu H, Simonyan K, Yang Y. Darts: Differentiable architecture search. ICLR 2019
>
> [4] Zhuge M, Wang W, Kirsch L, et al. Gptswarm: Language agents as optimizable graphs. ICML 2024.

---

> ### Author Response · Authors · 2025-11-25
>
> (Cont.)
>
> > **W2.** Baselines: The paper should include comparisons with more recent and relevant reasoning models to strengthen its empirical claims.
>
> **A2.** We thank the reviewer for suggestion about including more recent and relevant baselines. This work focuses on meta reasoning method, a technique that enables LLM aware of their reasoning process to improve reasoning ability at meta level. Meta reasoning methods can deal with some issues in LLM reasoning process such as getting stuck in a loop or deviating from the original goal. It is conceptually orthogonal to other technical routes for LLM reasoning such as Prompt Engineering, RAG or RLVR.
>
> In scope of meta reasoning, we use several recent and relevant baselines in our experiment, such as MRP(2024)[1], rStar (2025)[2] , Meta Reasoner(2025)[3] and MaAS(2025)[4].
>
> We also follow the reviewer's suggestion to conduct additional experiments to include another recent relevant baseline, i.e. Buffer-of-Thought (BoT) (2024)[5]. BoT retrieves meta-level thought template from a meta buffer to guide reasoning process. We conduct the experiment on MATH500 and Olympiad dataset. The results are summarized as follows.
>
> ---
>
> | Qwen3-8B-Thinking | MATH500 | Olympiad |
> | ----------------- | ------- | -------- |
> | **BoT**           | 83.6    | 52.1     |
> | **AutoMR**        | 86.6    | 53.9     |
>
> ---
>
> We add reference to this work and include more detailed analysis in Appendix C.3.3 of the updated manuscript.
>
> [1] Gao P, Xie A, Mao S, et al. Meta reasoning for large language models. 2024.
>
> [2] Qi Z, Ma M, Xu J, et al. Mutual reasoning makes smaller llms stronger problem-solvers. ICLR 2025.
>
> [3] Sui Y, He Y, Cao T, et al. Meta-reasoner: Dynamic guidance for optimized inference-time reasoning in large language models. 2025.
>
> [4] Zhang G, Niu L, Fang J, et al. Multi-agent architecture search via agentic supernet. ICML 2025.
>
> [5] Yang L, Yu Z, Zhang T, et al. Buffer of thoughts: Thought-augmented reasoning with large language models. NeurIPS 2024.
>
>
>
> > **W3.** Analysis Depth: Additional analysis could enhance the interpretability and transparency of the framework, helping readers gain a deeper understanding of its internal mechanisms and behavior. (e.g., error analysis to understand the limitations)
>
> **A3.** We provide additional case study in Appendix C.3.5. and the results in Figure 7. We provide a successful case (Figure 7 left) to show how AutoMR improve LLM reasoning performance compared with naive CoT. We also provide a failed case (Figure 7 right) to show the limitation that we have observed.
>
> **Successful Case**: As shown in Figure 7 left, step 3 (marked with a red cross) is a error step that is often reached by both AutoMR and naive CoT. However, in naive CoT, LLM will usually follow this error step for following reasoning process and reaches the error result in multiple runs. However, AutoMR can sample "Reflection" strategy for edge (3, 4) as shown in Figure 7 and find out this error to fix it in step 4 (marked with a green tick). Finally AutoMR can reach the correct answer.
>
> **Failed Case**: As shown in Figure 7 right, although AutoMR prompts LLM to recall related knowledge about the option A, LLM cannot recall comprehensive knowledge about option A, thus making error when judging correction of option A. Finally, LLM's output is wrong. This case shows that similar to many current LLM reasoning methods including RL, AutoMR can improve LLM reasoning performance but it may not be able to break through the ability boundary of base LLM.
>
> We add this additional case study and more detailed analysis in updated Appendix C.3.5.

---

> ### Author Response · Authors · 2025-11-25
>
> (Cont.)
>
> > **Q1.** The meta-reasoning behaviors summarized in the paper are derived from existing studies on LLM reasoning. Are these behaviors comprehensive enough? Have the authors considered incorporating other types of reasoning behaviors, perhaps inspired by cognitive or psychological perspectives?
>
> **A4.** Thanks for the suggestions about getting inspiration from cognitive or psychological science!
>
> **Scope and Grounding.** Our current meta-reasoning behavior set was distilled from behaviors most commonly reported in recent LLM cognition/reasoning studies (e.g., exploration and reflection), and is aligned with contemporary analyses of self-improving reasoners and long chain-of-thought practices [1,2].
>
> **On “Comprehensive.** Cognitive science suggests that meta-reasoning behaviors span a broad space and that no finite, universally optimal set exists across all tasks and domains [3,4]. Consequently, the behavior set should be adapted to the task distribution and evaluation goals.
>
> **Empirical Adequacy.** The behavior set used in AutoMR already yields strong results on our benchmarks, indicating that it is sufficiently expressive for the tasks evaluated.
>
> **Extensibility by Design.** Our framework treats behaviors as modular strategy types; it is straightforward to add, ablate, or specialize behaviors for new tasks. If future evidence shows that additional cognitively inspired behaviors (e.g., confidence-weighted verification, planning–execution decoupling) improve performance for a given domain, they can be incorporated with minimal changes to the search space.
>
>
> [1] Gandhi K, Chakravarthy A, Singh A, et al. Cognitive behaviors that enable self-improving reasoners, or, four habits of highly effective stars. COLM 2025.
>
> [2] Chen Q, Qin L, Liu J, et al. Towards reasoning era: A survey of long chain-of-thought for reasoning large language models. 2025
>
> [3] Flavell, J. H. Metacognition and cognitive monitoring: A new area of cognitive-developmental inquiry. 1979
>
> [4] Marion Rouault, et al. Human metacognition across domains: insights from individual differences and neuroimaging. 2018

---

> ### Author Response · Authors · 2025-11-26
>
> Dear reviewer,
>
> I hope this message finds you well. As the discussion deadline approaches next week, we would like to confirm that our replies have resolved your questions. If there are any remaining issues or additional feedback you would like to share, please feel free to let us know. Your insights have been highly valuable throughout this exchange.
>
> Thank you for the thoughtful review and your time.

---

> > ### Comment · Reviewer_SgMk · 2025-11-26
> >
> > Thanks for your response. I would suggest add the design motivation concisely in your revision. I will raise my score.

---

> > > ### Author Response · Authors · 2025-11-27
> > >
> > > Thank you for your sincere suggestion.
> > >
> > > We have updated the PDF to illustrate design motivation in Section 1 and Section 3.2.1 concisely. We also add Appendix A.5 to summarize the motivation in detail. We mark these text as blue.

---

### Official Review · Reviewer_SpuB · 2025-11-03

**Soundness:** 3
**Presentation:** 3
**Contribution:** 3
**Rating:** 6
**Confidence:** 2

**Summary:**

The paper purposes AutoMR, a meta-reasoning framework that treats the reasoning skeleton as a single-source, edge-heterogeneous DAG. The search space uses a strategy set plus a zero edge, enabling rich inter-step dependencies that prior fixed skeletons miss. Trained via policy-gradient over the sampler, AutoMR consistently outperforms CoT, Meta-Reasoner, rStar, etc. across math QA and MMLU-Pro subsets and exhibits superior token-scaling efficiency.

**Strengths:**

1. The paper reframes meta-reasoning as a creative, NAS-style control layer interleaved with the LLM’s ongoing reasoning rather than fixed upfront.
2. The unified DAG view plus inference-time search offers a general recipe for meta-control that can envelop many existing meta-reasoning templates.
3. The paper cleanly motivates the limitations of fixed skeletons, then walks through the search space and dynamic sampler with a concrete algorithmic presentation, which makes it easy to follow.

**Weaknesses:**

Results are on 3B instruction models (Qwen2.5-3B-Inst, LLaMA-3.2-3B-Inst) with a 1024-token budget for all methods. It remains unclear if AutoMR’s gains persist with stronger models or longer budgets typical in practice.

**Questions:**

1. When a node has multiple incoming edges with possibly different strategies, it is unclear how instructions are composed into one prompt for generating $c_i$​. (line 249-250)
2. Why not use Graph of thoughts as a baseline?
3. Some typos e.g. “dose” (line 161) “AutoTTS” (Table 1)

---

> ### Author Response · Authors · 2025-11-25
>
> We appreciate the reviewer’s time and valuable observations. Below is response to your proposed weakness and question.
>
> > **W1.** Results are on 3B instruction models (Qwen2.5-3B-Inst, LLaMA-3.2-3B-Inst) with a 1024-token budget for all methods. It remains unclear if AutoMR’s gains persist with stronger models or longer budgets typical in practice.
>
> **A1.** We conduct additional experiments based on larger LLM, including Qwen3-8B and Qwen3-14B, with larger token budget 8192.
> The performance of AutoMR and the baselines (Qwen3 thinking mode and two meta reasoning methods that performs well in Table 1) are summarized as follows:
>
> ---
>
> | Qwen3-8B-thinking | MATH500  | Olympiad |
> | ----------------- | -------- | -------- |
> | Base              | 80.8     | 47.0     |
> | rStar             | 83.4     | 48.9     |
> | Meta Reasoner     | 81.6     | 49.9     |
> | **AutoMR**        | **86.6** | **53.9** |
>
> ---
>
> | Qwen3-14B-thinking | MATH500  | Olympiad |
> | ------------------ | -------- | -------- |
> | Base               | 84.2     | 49.0     |
> | rStar              | 84.8     | 50.9     |
> | Meta Reasoner      | 85.6     | 50.4     |
> | **AutoMR**         | **87.2** | **54.5** |
>
> ---
>
> The results show that AutoMR also outperforms other baselines even when scaling to larger LLM and larger token budget.
>
> Additionally, in our original paper, we have already discussed the influence of token budget in **Section 4.3. Paragraph 1**, where we conduct experiments with different token budget (from 256 to 4096). The results are showed in **Figure 3**. Figure 3 shows that the performance of AutoMR and all baselines improves on the whole when token budget increases. But AutoMR consistently performs better than other baselines with different token budgets.
>
>
>
>
> > **Q1.** When a node has multiple incoming edges with possibly different strategies, it is unclear how instructions are composed into one prompt for generating $c_i$. (line 249-250)
>
> **A2.** When a node $n_i$ has multiple incoming edges with possibly different strategies, for each incoming edge $(j, i)$, we build the prompt for this edge by guiding LLM to attend to the specific node $n_j$ with strategy $s_{(j, i)}$. Then we concatenate prompts for all edges into one prompt to guide LLM to generate base reasoning content $c_i$for node $n_i$.
>
> For example, when node 2 has two incoming edges (0, 2) and (1, 2), we prompt LLM to attend to step represented by node 0 with strategy represented by edge (0, 2) and attend to step represented by node 1 with strategy represented by edge (1, 2).
>
> We add Appendix A.2 to show the prompt template and implementation details.
>
>
>
> > **Q2.** Why not use Graph of thoughts as a baseline?
>
> **A3.** Although GoT also use graph to represent LLM reasoning process, it is quite different from AutoMR and not applicable to the tasks in our experiments. We illustrate this point in detail as follows.
>
> 1. **Graph of AutoMR and GoT has different meanings and granularities**. GoT's graph has different meanings from AutoMR. For example, GoT follows the paradigm that decomposes the task into several subtasks and solve them respectively. Each node in its graph represents a solution to a subtask, rather than a reasoning step as in AutoMR. So AutoMR is more fine-grained. Moreover, the edge in GoT only represents dependency, while the edge also represents meta reasoning strategy in AutoMR.
> 2. **GoT requires user to design graph structure and configure it manually**. GoT in fact provides an interface for the user to design a static graph structure and related configuration manually but not build a thought graph itself automatically.
> 3. **GoT is not suitable to tasks in our experiments**. The graph indicating how to reason in GoT is static, this makes GoT only suitable to tasks where queries can be solved by fixed reasoning patterns (such as merge-based sorting for array with fixed length). This makes it struggle in tasks requiring diverse reasoning patterns used in our experiment (such as math).
>
> Despite the substantial differences from our approach, we still managed to compare against GoT as suggested.
> We implement a thought graph manually based on interface provided by GoT and evaluate it based on Qwen3-8B on MATH500 and Olympiad. We summarize the results as follows.
>
> ---
>
> | Qwen3-8B-thinking | MATH500 | Olympiad |
> | ----------------- | ------- | -------- |
> | Base              | 80.8    | 47.0     |
> | **GoT**           | 78.6    | 45.7     |
> | **AutoMR**        | 86.6    | 53.9     |
>
> ---
>
> The results show that GoT cannot work well for the tasks of interest in our experiment. We add reference to this work and include these results and analysis in the updated Appendix C.3.3.
>
>
>
>
> > **Q2.** Some typos e.g. “dose” (line 161) “AutoTTS” (Table 1)
>
> **A4.** We thank the reviewer for pointing out these typos. We fix these typos in the updated version.

---

### Official Review · Reviewer_iKDH · 2025-11-04

**Soundness:** 2
**Presentation:** 2
**Contribution:** 2
**Rating:** 4
**Confidence:** 4

**Summary:**

This paper introduces AutoMR, a framework that improves large language model (LLM) reasoning performance by automatically searching for query-aware meta-reasoning skeletons. It represents these skeletons as directed acyclic graphs (DAGs) and employs a dynamic sampling algorithm. Experiments demonstrate that AutoMR improves reasoning performance and efficiency compared to existing methods that rely on manually designed skeletons.

**Strengths:**

1. This paper introduces a new method that eliminate the need for traditional manual design.
2. The dynamic skeleton sampling algorithm enables the generated meta-reasoning skeletons to possess strong query-awareness and adaptability.

**Weaknesses:**

I have the following concerns. *If the authors could address my concerns during the rebuttal stage, I will consider raising my score.*
1. Despite the authors stating that 3B models were used for fair comparison, I am still curious about the potential performance gains when applying AutoMR to larger LLMs, especially given the method's reported low training cost and inference efficiency.
2. The performance improvement on knowledge-intensive tasks is notably more limited compared to thinking-intensive tasks, and the method's performance on knowledge-intensive tasks seems insufficient to demonstrate the method's advantages fully.
3. As the paper only evaluates the method on math Q&A and general multiple-choice tasks across different disciplines and difficulties, I am worried about the method's scalability and broad effectiveness across a wider range of diverse tasks. I look forward to the authors providing performance results on complex reasoning datasets, including, but not limited to, Game of 24 [1], BIG-Bench Hard (BBH) [2], and Python Programming Puzzles [3].
4. Buffer of Thoughts (BoT) [4] is a great thought-augmented reasoning approach, which utilizes a meta-buffer to store and retrieve high-level thought-templates distilled from various problem-solving processes. I'm very curious about the performance difference between this work and BoT.
5. There are several spelling errors in the paper, for instance, "an" in the Table 1 caption, "subet" on page 7, and "to to enhance" on page 8.

[1] Tree of thoughts: Deliberate problem solving with large language models. NeurIPS 2024.

[2] Challenging big-bench tasks and whether chain-of-thought can solve them. ACL 2023 Findings.

[3] Programming puzzles, in Thirty-fifth Conference on Neural Information Processing Systems Datasets and Benchmarks Track,
2021.

[4] Buffer of Thoughts: Thought-Augmented Reasoning with Large Language Models. NeurIPS 2024.

**Questions:**

Please see Weaknesses.

---

> ### Author Response · Authors · 2025-11-25
>
> We sincerely thank the reviewer for the thoughtful and constructive feedback.
>
> > **W1.** Despite the authors stating that 3B models were used for fair comparison, I am still curious about the potential performance gains when applying AutoMR to larger LLMs, especially given the method's reported low training cost and inference efficiency.
>
> **A1.** We conduct additional experiments to evaluate our proposed **AutoMR** based on larger long-cot LLMs (including Qwen3-8B and Qwen3-14B) with larger token budget (8192) on two datasets (MATH500 and Olympiad). The performance of AutoMR and the baselines (Qwen3 thinking mode and two meta reasoning methods that performs well in Table 1) are summarized as follows:
>
> ---
>
> | Qwen3-8B-thinking | MATH500  | Olympiad |
> | ----------------- | -------- | -------- |
> | Base              | 80.8     | 47.0     |
> | rStar             | 83.4     | 48.9     |
> | Meta Reasoner     | 81.6     | 49.9     |
> | **AutoMR**        | **86.6** | **53.9** |
>
> ---
>
> | Qwen3-14B-thinking | MATH500  | Olympiad |
> | - | -| -|
> | Base| 84.2     | 49.0     |
> | rStar | 84.8     | 50.9     |
> | Meta Reasoner| 85.6     | 50.4     |
> | **AutoMR**| **87.2** | **54.5** |
>
> ---
>
> These results show that **AutoMR** still achieves performance gains when scaling to larger LLM (i.e. 8B and 14B), compared with naive LLM thinking and previous meta reasoning methods, demonstrating the scaling potential of AutoMR.
>
> We include these results and analysis in Appendix C.3.1 of the updated manuscript.
>
>
> > **W2.** The performance improvement on knowledge-intensive tasks is notably more limited compared to thinking-intensive tasks, and the method's performance on knowledge-intensive tasks seems insufficient to demonstrate the method's advantages fully.
>
> **A2.** Both **Table 1** and **Figure 3** clearly show that AutoMR achieves substantial improvements over CoT and other baselines on both thinking-intensive math problems and relatively knowledge-intensive multiple-choice questions. In the updated version, we also added the blue numbers in Table 1 to highlight AutoMR’s gains over CoT.
>
> The key distinction between the two task types is discussed in Section 4.3 (Paragraph 1), where we analyze the influence of token-budget scaling. The results are shown in Figure 3.
>
> Figure 3 indicates that the performance of AutoMR and all baselines generally increases as the token budget grows. However, the improvement rate on knowledge-intensive multiple-choice tasks is noticeably slower than on thinking-intensive math tasks. Importantly, AutoMR consistently maintains its advantage over all baselines at every budget. This slower improvement trend aligns with recent findings [1], suggesting that it may be a characteristic of knowledge-intensive tasks rather than a limitation of specific reasoning methods.
>
> [1] James Xu Zhao, Bryan Hooi, and See-Kiong Ng. Test-time scaling in reasoning models is not effective for knowledge-intensive tasks yet.
>
>
> > **W3.** As the paper only evaluates the method on math Q&A and general multiple-choice tasks across different disciplines and difficulties, I am worried about the method's scalability and broad effectiveness across a wider range of diverse tasks. I look forward to the authors providing performance results on complex reasoning datasets, including, but not limited to, Game of 24 [1], BIG-Bench Hard (BBH) [2], and Python Programming Puzzles [3].
>
> **A3.** We conduct additional experiments based on Qwen3-8B on three given complex reasoning, including Game of 24, BIG-Bench Hard (BBH) and Python Programming Puzzles (P3).
>
> We summarize the results as follows:
>
> ---
>
> | Qwen3-8B-thinking | Game of 24 | BBH      | P3       |
> | -- | -| -------- | -------- |
> | Base              | 73.0       | 80.2     | 75.0     |
> | rStar             | 75.0       | 83.9     | 76.8     |
> | Meta Reasoner     | 77.0       | 80.6     | 79.6     |
> | **AutoMR**        | **79.0**   | **85.6** | **84.4** |
>
> ---
>
> The results show that AutoMR still achieves better overall performance on these three datasets. Specifically, the advantage is most pronounced on P3, followed by BBH, while the gain on Game of 24 is relatively small. We argue that this difference arises because these datasets require fundamentally different reasoning patterns.
>
> * **Game of 24 shows similar reasoning strategies**. On Game of 24, all methods tend to use similar strategies: enumerating combinations of the four given numbers with operations (+, -, *, /) and checking them one by one. AutoMR also discovers this meta-reasoning skeleton, leading to comparable performance across methods.
>
> * **P3 requires much more complex meta reasoning strategies**. P3 demands substantially more complex strategies, such as exploring alternative solutions or reflecting on intermediate attempts. In this setting, the DAG-based meta-reasoning skeleton proposed by AutoMR is far more effective than the baselines.
>
> We include these results and analysis in Appendix C.3.2 of the updated manuscript.

---

> ### Author Response · Authors · 2025-11-25
>
> (Cont.)
> > **W4.** Buffer of Thoughts (BoT) [4] is a great thought-augmented reasoning approach, which utilizes a meta-buffer to store and retrieve high-level thought-templates distilled from various problem-solving processes. I'm very curious about the performance difference between this work and BoT.
>
> **A4.** Thank you for pointing out this relevant work.
> We conducted additional experiments to evaluate the performance difference between AutoMR and BoT. The experiments use Qwen3-8B on five datasets: MATH, Olympiad, Game of 24, BBH, and P3. Since the open-source code of BoT provides some templates to initialize meta buffer as well as update it dynamically by prompting LLM summarize new templates, we also design a variant named **BoT-EmptyInit** which removes the provided templates but retains the dynamic update process for a fair comparison. Because the BoT code does not include templates for P3 and most BBH subsets, BoT is equivalent to BoT-EmptyInit on these datasets.
> We summarize the results as follows.
>
> ---
>
> | Qwen3-8B-Thinking | MATH500  | Olympiad | Game of 24 | BBH      | P3       |
> | ----------------- | -------- | -------- | ---------- | -------- | -------- |
> | **BoT-EmptyInit** | 81.0     | 47.4     | 76.0       | **85.7** | 80.6     |
> | **BoT**           | 83.6     | 52.1     | **94.0**   | **85.7** | 80.6     |
> | **AutoMR**        | **86.6** | **53.9** | 79.0       | 85.6     | **84.4** |
>
> ---
>
> The results show that AutoMR performs better than BoT on MATH500, Olympiad and PPP. The performance is similar on BBH and BoT outperforms AutoMR on Game of 24. We explain such difference as follows:
>
> * **Characteristic of Thought Template**: Thought template is highly effective when it matches given query. However, thought template is hard to generalize to different queries when they have relatively large gap. Mismatching may lead to template ineffective or even mislead reasoning process.
> * **Game of 24 has solution that can generalize well**: The combination of four numbers and +, -, *, / is finite, so there is a general template by checking all combinations exhaustively. Such task is quite suitable for thought template. BoT provides python code as thought template to implement such exhaustive process. Therefore, BoT performs much better on Game of 24.
> * **Queries from MATH500, Olympiad and P3 are much more diverse**: This makes the templates provided by BoT source code or summarized by LLM hard to generalize to different queries. Therefore, AutoMR performs better on these three datasets.
>
> We add reference to this work and include these results and analysis in Appendix C.3.3 of the updated manuscript.
>
>
> > **W5.** There are several spelling errors in the paper, for instance, "an" in the Table 1 caption, "subet" on page 7, and "to to enhance" on page 8.
>
> **A5.** Thank you for pointing out these spelling errors. We fix these errors in the updated version.

---

> ### Author Response · Authors · 2025-11-26
>
> Dear reviewer,
>
> As we are approaching the end of the discussion period next week, we would like to check whether there are any remaining points you wish us to elaborate on. Please let us know if additional clarification would be helpful. Your perspective has been highly valuable throughout this process.
>
> Thank you for your time and consideration.

---

> ### Comment · Reviewer_iKDH · 2025-11-27
> **Thank you for the detailed response.**
>
> Thank you for the detailed response, but I still have some reservations.
>
> Your experimental results show that BoT performs very well on tasks with a small exploration space and remains reasonably effective on tasks requiring more exploration. In contrast, AutoMR provides only limited improvement over BoT on tasks with a large exploration space, while performing worse on tasks with a small exploration space. This raises concerns about the scalability and generalizability of the proposed method.

---

> ### Author Response · Authors · 2025-11-28
>
> Thank you for your timely comment about our additional experiment on BoT.
>
> But we need to illustrate that the comparison between BoT and AutoMR in A4 is not fair, we explain this with Game of 24 as follows.
>
> **Game of 24 has a brute-force general solution that can be implemented by code easily.** Game of 24 dose not certainly require less exploration, but as we have illustrated in A4, it has a general solution that can solve all queries in this task with 100% accuracy, just by brute-force checking all combinations of 4 given numbers and +,-,*,/, until a correct combination is found. Such brute-force process can be implemented easily by code.
>
> **BoT uses such brute-force python code as the prompt rather than let LLM to distill such code during reasoning process itself, leading to unfair comparison.** BoT instructs LLM to fill the four given numbers into the code and then BoT executes the code to get the final answer. It is quite easy for current LLM to just fill 4 numbers into the code, therefore BoT achieves 97% accuracy, but it is a quite unfair comparison. In fact, such implementation is meaningless because for any task that can be solved by code, we can just use the code directly to solve it. Using code directly is much more efficient than using it to prompt LLM and can achieve 100% accuracy.
>
> **Under fair comparison, AutoMR performs much better than BoT.** For fair comparison, the thought templates should not be provided in advance (i.e. BoT-EmptyInit in A4) but the LLM need to distill thought templates itself on the dataset used for searching meta reasoning skeleton in AutoMR. We summarize the results as follows. This comparison is fair and BoT's performance on Game of 24 reduces to 76%, lower than that of AutoMR. Moreover, AutoMR outperforms BoT significantly on MATH500, Olympiad and P3.
>
> ---
>
> | Qwen3-8B-Thinking | MATH500          | Olympiad         | Game of 24       | BBH          | P3               |
> | ----------------- | ---------------- | ---------------- | ---------------- | ------------ | ---------------- |
> | **BoT**           | 81.0             | 47.4             | 76.0             | **85.7**     | 80.6             |
> | **AutoMR**        | **86.6 (+5.6%)** | **53.9 (+6.5%)** | **79.0 (+3.0%)** | 85.6 (-0.1%) | **84.4 (+3.8%)** |
>
> ---
>
> **AutoMR is orthogonal to BoT.** BoT provides thought templates to guide LLM before reasoning process starts, while AutoMR incorporates meta reasoning behaviors along with reasoning process.
>
> **When compared with other baselines fairly, AutoMR achieves consistent performance improvement.** As shown in A3, among the methods that do not use brute-force code to prompt LLM, AutoMR performs the best on Game of 24, BBH and P3, demonstrating the generalizability of AutoMR.

---

### Meta-Review · Area_Chair_zdSP · 2026-01-06

**Summary:**

This paper proposes AutoMR, a meta-reasoning framework that models reasoning skeletons as query-aware directed acyclic graphs and automatically searches them at inference time. Unlike prior work with manually designed or static reasoning structures, AutoMR dynamically interleaves skeleton construction with the evolving reasoning context, enabling richer dependencies such as branching, reflection, and verification. The method is trained with lightweight reinforcement learning. Reviewers are concerned about the evaluation, which does not systematically demonstrate the benefits of the designed system.

**Reviewer Concerns:**

Reviewer iKDH: The reviewer questioned the scalability and generalizability of AutoMR, particularly in comparison with Buffer-of-Thought, noting weaker performance on tasks with small, brute-force-friendly solution spaces such as Game of 24. They were concerned that AutoMR’s advantages may diminish relative to template-based methods and asked whether the method truly generalizes across diverse reasoning regimes.

Reviewer SpuB: The reviewer requested clarification on prompt composition when multiple incoming edges exist and questioned the relevance of Graph-of-Thought as a baseline.

Reviewer SgMk: The reviewer requested deeper analysis of internal mechanisms and limitations, as well as broader baseline coverage to strengthen empirical claims.

Reviewer CbsY: The reviewer raised concerns about limited evaluation on long-CoT and larger models, the practical niche of AutoMR relative to parameter-efficient fine-tuning methods such as LoRA, and conceptual overlap with Graph-of-Thought. Additional questions focused on the choice and sufficiency of meta-reasoning strategies, training data efficiency, and some issues with presentation and organization.

**Reviewer Scores:**

The main problem of the paper is the evaluation. Reviewers requested several additional experiments and authors provided a few more cases, which in my opinion is not sufficient. Now, the whole evaluation is still not systematic, namely, there is no strong evidence or signal to confidently say that the proposed method would be better. On the contrary, it is worse than some alternatives, e.g., buffer-of-thoughts, as acknowledged by the experiments. The problem is the paper could not give a convincing reason why some would favor the proposed method.

---

### Decision · Program_Chairs · 2026-01-26

Reject